# REAL-TIME DESIGN OF ARCHITECTURAL STRUCTURES WITH DIFFERENTIABLE MECHANICS AND NEURAL NETWORKS

**Rafael Pastrana,**[1] **Eder Medina,**[2] **Isabel M. de Oliveira,**[3]
**Sigrid Adriaenssens,**[3] **Ryan P. Adams**[2]
[1] Architecture    [2] Computer Science    [3] Civil and Environmental Engineering
Princeton University
`{arpastrana,em2368,imdo,sadriaen,rpa}@princeton.edu`

## ABSTRACT

Designing mechanically efficient geometry for architectural structures like shells, towers, and bridges, is an expensive iterative process. Existing techniques for solving such inverse problems rely on traditional optimization methods, which are slow and computationally expensive, limiting iteration speed and design exploration. Neural networks would seem to offer a solution via data-driven amortized optimization, but they often require extensive fine-tuning and cannot ensure that important design criteria, such as mechanical integrity, are met. In this work, we combine neural networks with a differentiable mechanics simulator to develop a model that accelerates the solution of shape approximation problems for architectural structures represented as bar systems. This model explicitly guarantees compliance with mechanical constraints while generating designs that closely match target geometries. We validate our approach in two tasks, the design of masonry shells and cable-net towers. Our model achieves better accuracy and generalization than fully neural alternatives, and comparable accuracy to direct optimization but in real time, enabling fast and reliable design exploration. We further demonstrate its advantages by integrating it into 3D modeling software and fabricating a physical prototype. Our work opens up new opportunities for accelerated mechanical design enhanced by neural networks for the built environment.

## 1  INTRODUCTION

Mechanical efficiency is required for architectural structures to span hundreds of meters under extreme loads safely with low material volume. An efficient structure sustains loads with small physical element sizes compared to its span, such as thin bars or slender plates, thus minimizing its material footprint. Additionally, shells, towers, and bridges—examples of such systems—must comply with geometric constraints arising from architecture and fabrication requirements to become feasible structures in the built environment. Designing shapes for such long-span structures, which must fulfill mechanical efficiency and geometric constraints, is a complex task requiring substantial domain expertise and human effort. Our goal is to use machine learning to accelerate this challenging task without compromising safety-critical aspects of the design.

One way to approach this problem is to start from the mechanical standpoint, employing a specialized mechanical model that directly computes efficient geometry for structures modeled as a bar systems (Bletzinger and Ramm, 2001; Bletzinger et al., 2005). Unlike standard, finite element-based mechanical analysis, where one first defines the structure's geometry and then obtains its internal forces, these specialized models—known as *form-finding methods* in structural engineering (Veenendaal and Block, 2012; Adriaenssens et al., 2014)—reverse the relationship between geometry and force to produce mechanically efficient shapes in a single forward solve (Shin et al., 2016). As a result, these methods have been successfully applied to design landmark structures with thickness-to-span ratios up to 1:70 (less than that of an eggshell), across a wide palette of materials, including stainless steel (Schlaich, 2018), reinforced concrete (Isler, 1994), and stone (Block et al., 2017).

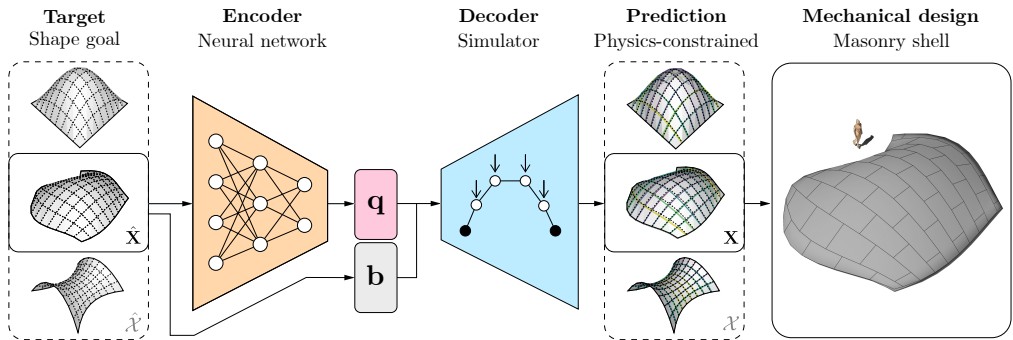

Figure 1: Architecture of our model to amortize the generation of mechanically efficient geometry. A neural network first maps a target shape $\hat{\mathbf{X}}$ sampled from a family of shapes $\hat{\mathcal{X}}$ to a stiffness space $\mathbf{q}$. The stiffnesses, in tandem with the boundary conditions $\mathbf{b}$, are then decoded by a mechanical simulator into a physics-constrained shape $\mathbf{X}$ that approximates the target. The predicted shape can be used as the base geometry to design efficient structures like gridshells or masonry shells.

While utilizing a specialized model can lead to efficient designs, it is difficult to guide solutions toward particular geometries as the designer only has explicit control of the mechanical behavior and not the shape. To solve this inverse problem, form-finding is complemented with optimization algorithms to find internal force states that satisfy geometric goals (Panozzo et al., 2013; Maia Avelino et al., 2021). For a designer, however, it is cumbersome to perform a time-consuming and computationally intensive optimization when exploring shapes. Practical design requires the evaluation of several target shapes to align with geometric desiderata, multiplying computational effort as each shape requires its own optimization. Therefore, using specialized mechanical models and optimization together is effective but inefficient in practice, where shape variety and real-time feedback are essential.

Neural networks (NNs) can accelerate the mechanical design of architectural structures with data-driven surrogate models that amortize inverse problems, enabling more responsive design tools. Recent applications include the design of tall buildings (Chang and Cheng, 2020), truss lattices (Bastek et al., 2022), and reticulated shells (Tam, 2023). Nevertheless, these purely data-driven examples require learning representations of the inverse problem and the underlying physics at the same time. Even with physics-informed neural networks (PINNs) (Raissi et al., 2019b; Karniadakis et al., 2021; Lu et al., 2021; Bastek and Kochmann, 2023) trained with sophisticated loss balancing schemes (Bischof and Kraus, 2021; Wang et al., 2022), there is **no guarantee of mechanical integrity** in their predictions. Here, we define integrity as the accuracy in predicting the mechanical response of a structure by respecting physics laws. Assurance of mechanical integrity is a foundational tenet in structural engineering, where poor predictions might lead to catastrophic collapse and the loss of human lives. In contrast, mechanical simulators in structural engineering have been developed for decades and offer a principled and interpretable way to represent the physics of long-span structures. A hybrid solution would seem ideal, in which neural network amortization is integrated with differentiable physics models (Belbute-Peres et al., 2020; Um et al., 2021; Thuerey et al., 2022; Oktay et al., 2023) to construct a class of machine learning models that shift the current paradigm from a physics-informed to a physics-in-the-loop approach for safety-critical applications.

In this paper, we develop a neural surrogate model that couples a neural network with a differentiable mechanics simulator to enable the solution of shape approximation problems for architectural structures in real time (Figure 1). The coupled model offers advantages over direct gradient-based optimization and current fully neural alternatives for interactive mechanical design. We evaluate our method in two design tasks of increasing complexity: masonry shells and cable-net towers. Our contributions are threefold. First, we demonstrate that **our model generates mechanically sound predictions** at higher accuracy than NNs and PINNs of similar architecture. Our model exhibits better generalization performance than an equivalent PINN in the masonry shells task. Second, we show that our model reaches comparable accuracy to optimization, but it is up to four orders of magnitude faster. In the cable-nets task, our model provides robust initialization for direct optimization, outperforming the designs generated by optimization initialized with human domain expertise. Third, we showcase the application of a maturing machine learning technique (i.e., coupling learnable and analytical components in the same architecture) to a new, high-impact domain for physical design (i.e., architectural structures). To illustrate its practical impact, we deploy our trained model in a 3D modeling program to design a shell and then fabricate a physical prototype of the predicted geometry. Source code is available at `https://github.com/princetonlips/neural_fdm`.

## 2 PHYSICS-CONSTRAINED NEURAL FORM DISCOVERY

Our goal is to generate mechanically efficient shapes for architectural structures that approximate target geometries in real time while maintaining mechanical integrity. The challenge is that, because mechanical efficiency is important in architectural structures, it is necessary to reason about designs from the point of view of force balance; but the resulting geometries are a nontrivial function of their mechanical behavior. Thus we seek to use machine learning to efficiently invert this function to generate target designs without compromising their integrity.

### 2.1 COMPUTING EFFICIENT GEOMETRY

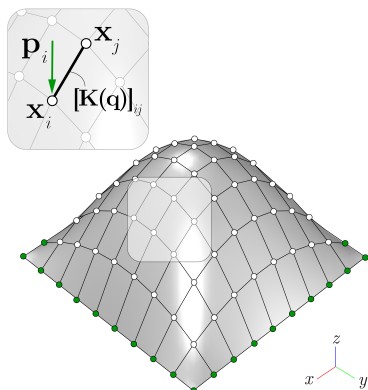

We focus on structures modeled as pin-jointed bar systems of $N$ nodes connected by $M$ bars (Figure 2). Each node experiences an external load vector (e.g., self-weight or wind load) and some nodes are constrained to fixed positions (e.g., terrain and anchors). These are the structure's boundary conditions $\mathbf{b} \in \mathbb{R}^L$. After picking bar stiffnesses $\mathbf{q} \in \mathbb{R}^M$, the goal of form-finding is to identify the positions of the nodes $\mathbf{X} = (\mathbf{x}_1, \ldots, \mathbf{x}_N) \in \mathbb{R}^{N \times 3}$ such that there is no net residual force on the structure.

Figure 2: A bar system. In the callout, the stiffness component $[\mathbf{K}(\mathbf{q})]_{ij}$ is indicated at a bar connecting nodes with positions $\mathbf{x}_i$ and $\mathbf{x}_j$. A load $\mathbf{p}_i$ is applied at $\mathbf{x}_i$. In this system, the nodes on the perimeter are fixed.

An efficient structure is one whose loaded configuration is bending- and torsion-free, therefore reducing the material volume required to resist external loads. This configuration minimizes the structure's strain energy by reducing the contribution of such components, letting a structure sustain loads mainly under tensile and compressive axial forces. We can satisfy this property if the shape of a structure modeled as a bar system is in equilibrium. To arrive at equilibrium, the residual force vector $\mathbf{r}_i \in \mathbb{R}^3$ at a free node with position $\mathbf{x}_i$ must be zero. The function $\mathbf{r}_i(\mathbf{X}; \mathbf{q})$ quantifies the difference between the load $\mathbf{p}_i \in \mathbb{R}^3$ applied to a node and the internal forces from the bars connected to it, for given positions, stiffness values, and boundary conditions:

$$\mathbf{r}_i(\mathbf{X}; \mathbf{q}) = \sum_{j \in \mathcal{N}(i)} [\mathbf{K}(\mathbf{q})]_{ij}(\mathbf{x}_j - \mathbf{x}_i) - \mathbf{p}_i(\mathbf{x}_i) \tag{1}$$

where $\mathcal{N}(i)$ are the node neighbors, and $\mathbf{K}(\mathbf{q}) \in \mathbb{R}^{N \times N}$ is the stiffness matrix as a function of $\mathbf{q}$. The restriction that there is no residual force can be framed as a constraint in which all of the $\mathbf{r}_i(\mathbf{X}; \mathbf{q}) = \mathbf{0}$ at the free nodes. Although this constraint can be solved with mechanical simulators like form-finding methods (Adriaenssens et al., 2014), the resulting map from stiffness to geometry, which we denote $\mathbf{X}(\mathbf{q})$, is implicit and nonlinear, and difficult to reason about directly.

### 2.2 DIRECT OPTIMIZATION FOR TARGET SHAPES

Even though form-finding methods have the appealing property of promoting mechanical efficiency, they do not allow a designer to directly target particular geometries. Moreover, not all arbitrary geometries are even compatible with mechanical efficiency. If a designer has a target shape $\hat{\mathbf{X}}$, they wish to solve the following optimization problem with respect to $\mathbf{q}$ to approximate $\hat{\mathbf{X}}$ with $\mathbf{X}$:

$$\mathbf{q}^\star = \arg\min_{\mathbf{q} \in \mathbb{R}^M} \mathcal{L}_{\text{shape}}(\mathbf{q}) \qquad \text{where} \qquad \mathcal{L}_{\text{shape}}(\mathbf{q}) = \sum_{i=1}^{N} \sum_{d=1}^{3} \|[\mathbf{X}(\mathbf{q})]_{i,d} - [\hat{\mathbf{X}}]_{i,d}\|^p \tag{2}$$

and $p > 0$. We call $\mathcal{L}_{\text{shape}}$ the *shape loss*. This objective, which we refer to as *direct optimization*, identifies bar stiffness values $\mathbf{q}$ that generate shapes close to the designer's intent in an $\ell_p$ sense while maintaining net zero force balance. Note that the optimization setup does not contain any information about the set of physically valid forms and it is simply driven by the minimization of the pointwise difference between shapes. Conventionally, this nonlinear optimization problem is solved numerically in the inner loop of a design process, but that is slow and computationally costly (Figure 3).

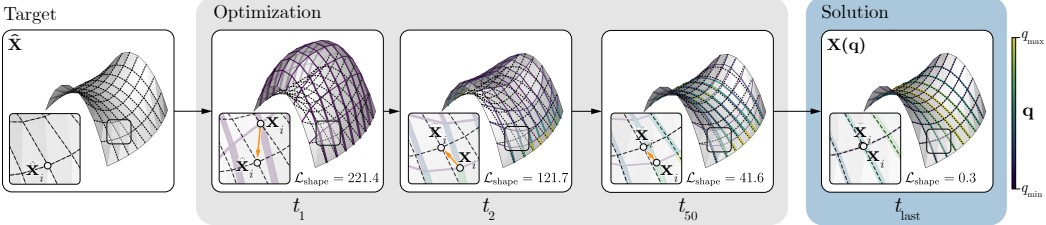

Figure 3: Shape matching. To generate a mechanically efficient shape $\mathbf{X}(\mathbf{q})$ that approximates an arbitrary target $\hat{\mathbf{X}}$, traditional methods like direct optimization find bar stiffnesses $\mathbf{q}$ that minimize the shape loss $\mathcal{L}_{\text{shape}}$, but only after several iterations $t$. Our model amortizes this computationally taxing process during inference while guaranteeing physics, enabling real-time and sound design.

## 2.3 AMORTIZED SHAPE MATCHING

Rather than performing many costly optimizations within a design loop, we use a machine learning model to amortize the solution over a family of target shapes $\hat{\mathcal{X}}$. A neural network takes as input the designer's intent $\hat{\mathbf{X}}$, and outputs the stiffness values $\mathbf{q}$ such that $\mathbf{X}(\mathbf{q}) \approx \hat{\mathbf{X}}$. Our model architecture resembles an autoencoder (Figure 1). First, a neural encoder $\mathcal{E}_\phi$ maps the target shape $\hat{\mathbf{X}}$ into $\mathbf{q}$. Then, a mechanical simulator *decodes* the associated shape $\mathbf{X}(\mathbf{q})$, subject to the boundary conditions $\mathbf{b}$. The key property of this construction is that **the resulting shape is mechanically sound even if the neural network is inaccurate**; the failure mode is not a lack of mechanical integrity, but a shape that does not match the target very well.

**Neural encoder**    The encoder is a neural network with learnable parameters $\phi$ that ingests the targets and projects them into a stiffness space, $\mathcal{E}_\phi : \mathbb{R}^{N \times 3} \to \mathbb{R}^M$. For simplicity, we cast our problem as a point-wise matching task and use a multilayer perceptron (MLP) as the encoder, although we are not restricted to that. Regardless of the neural network specification, the output representation must be strictly positive. This is necessary for compatibility with our mechanical simulator to avoid null stiffness values that bear limited physical meaning in our representation of an architectural structure. We satisfy this requirement by applying a strictly positive nonlinearity to the last layer of the encoder.

One of the advantages of our mechanical simulator is that it enables us to prescribe tensile or compressive bar forces *a priori*. As a result, rather than making these force directions a learnable feature, we build this bias into the encoder architecture by scaling the strictly positive embedding of the last layer by a force direction vector $\mathbf{s} \in \mathbb{R}^M$. The scaling factors $s \in \{-1, 1\}$ indicate the direction of the internal axial force of every bar: a negative factor $s$ prescribes a compressive force, and a positive factor, a tensile force. Our encoder thus calculates the bar stiffness values $\mathbf{q}$ as:

$$\mathbf{q} = \mathbf{s} \odot (\sigma(\mathbf{h}) + \tau) \tag{3}$$

where $\sigma$ is the strictly positive nonlinearity, $\mathbf{h}$ is the encoder's last layer embedding, $\tau \geq 0$ is a fixed scalar shift that specifies a minimum stiffness value for the entries of $\mathbf{q}$ (akin to a box constraint in numerical optimization), and $\odot$ indicates the element-wise product. Setting a lower bound on the stiffness values with $\tau$ guides the learning process towards particular solutions since the map from $\mathbf{q}$ to $\mathbf{X}$ is not unique (Van Mele et al., 2012), and provides numerical stability when amortizing over structures with complex force distributions (Section 4.2).

**Mechanical decoder**    The decoder gives the latent space of our model a physical meaning since it represents the inputs of a mechanical simulator. To fulfill equilibrium in a pin-jointed bar system, the relationship between the stiffness values $\mathbf{q}$, the shape $\mathbf{X}$, and the loads $\mathbf{P} \in \mathbb{R}^{N \times 3}$ must satisfy $\mathbf{K}(\mathbf{q})\,\mathbf{X} - \mathbf{P}(\mathbf{X}) = \mathbf{0}$. Solving this equation for a bar system using finite element-based simulators (Xue et al., 2023; Wu, 2023) is possible, but requires second-order methods. Controlling the force signs adds numerical complexity, but it is a desirable property to design structures built from tailored materials that are strong only in tension or compression (e.g., masonry blocks or steel cables). Here, we utilize the force density method as our simulator (Pastrana et al., 2023b). Appendix A offers an extended description, but at a high level, the simulator is a form-finding method that linearizes the equilibrium constraint by assuming independence between stiffness, geometry, and loads. This reduces the computation of $\mathbf{X}(\mathbf{q}) : \mathbb{R}^M \to \mathbb{R}^{N \times 3}$ with target force signs to a linear solve:

$$\mathbf{X}(\mathbf{q}) = \mathbf{K}(\mathbf{q})^{-1} \mathbf{P} \tag{4}$$

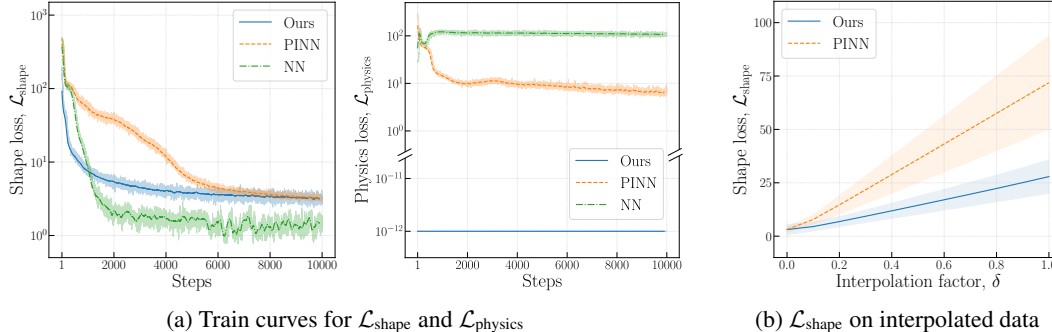

(a) Train curves for $\mathcal{L}_{\text{shape}}$ and $\mathcal{L}_{\text{physics}}$          (b) $\mathcal{L}_{\text{shape}}$ on interpolated data

Figure 4: Loss curves of the shell design task. (a) Our model learns a meaningful representation that minimizes the shape loss $\mathcal{L}_{\text{shape}}$ while fully satisfying the mechanics of compression-only shells, as $\mathcal{L}_{\text{physics}}$ is zero within numerical precision throughout training. (b) Shape loss of our model and the PINN baseline on test data interpolated between doubly-symmetric ($\delta = 0$) and asymmetric ($\delta = 1$) geometries. Our model's accuracy decays at a lower rate than the PINN's accuracy.

**Training** To amortize the shape-matching problem, we look for model parameters $\phi^{\star}$ that minimize the expected value of the shape loss:

$$\phi^{\star} = \arg\min_{\phi} \mathbb{E}_{\hat{\mathbf{X}} \sim \hat{\mathcal{X}}} \left[ \sum_{i=1}^{N} \sum_{d=1}^{3} \|[\mathbf{X}(\mathcal{E}_{\phi}(\hat{\mathbf{X}}))]_{i,d} - [\hat{\mathbf{X}}]_{i,d}\|^{p} \right] \tag{5}$$

We train our model via first-order stochastic gradient descent, averaging the loss values over batches of size $B$ at each training step. Appendix B provides training specifications. We generate training data by sampling batches of target shapes $\hat{\mathbf{X}}$ from a task-specific family of shapes $\hat{\mathcal{X}}$ parametrized by a probability distribution (Section 4). Our model can be trained end-to-end because the encoder and decoder are both implemented in a differentiable programming environment (Bradbury et al., 2018). As a result, reverse-mode automatic differentiation can seamlessly backpropagate the physics-based gradients that tune the neural network parameters.

## 3 EVALUATION

We evaluate model performance by measuring the shape loss $\mathcal{L}_{\text{shape}}$ and the inference wall time on a test dataset of target shapes. We compare the performance of our model to that of three other baselines: a fully neural model (NN), a fully neural model augmented with a physics-informed loss (PINN), and direct optimization.

The fully neural approaches replace the differentiable simulator in our model with a learnable decoder mirroring the encoder's architecture, with the addition of the boundary conditions $\mathbf{b}$ as inputs. The first baseline is trained to minimize the shape loss $\mathcal{L}_{\text{shape}}$ alone, highlighting that an information bottleneck is insufficient to generate physically plausible designs. The second baseline extends the first baseline by adding an explicit physics loss term:

$$\mathcal{L}_{\text{physics}} = \mathbb{E}_{\hat{\mathbf{X}} \sim \hat{\mathcal{X}}} \left[ \|\mathbf{R}(\mathbf{X}(\mathcal{E}_{\phi}(\hat{\mathbf{X}})))\|_{F} \right] \tag{6}$$

The physics loss measures the Frobenius norm of the residual forces $\mathbf{R} \in \mathbb{R}^{N_u \times 3}$ at the $N_u$ free nodes of a structure (see Appendix A). For a shape to be physically valid in our setup, $\mathcal{L}_{\text{physics}}$ must be zero within numerical precision (e.g., $1 \times 10^{-12}$). We reason that the additional term should provide a training signal to the encoder and decoder of the PINN so that they learn to solve the shape-matching task and the physics concurrently. The third baseline takes advantage of the differentiable mechanical simulator and optimizes $\mathbf{q}$ directly to minimize the shape loss via deterministic gradient-based optimization on a per-shape basis.

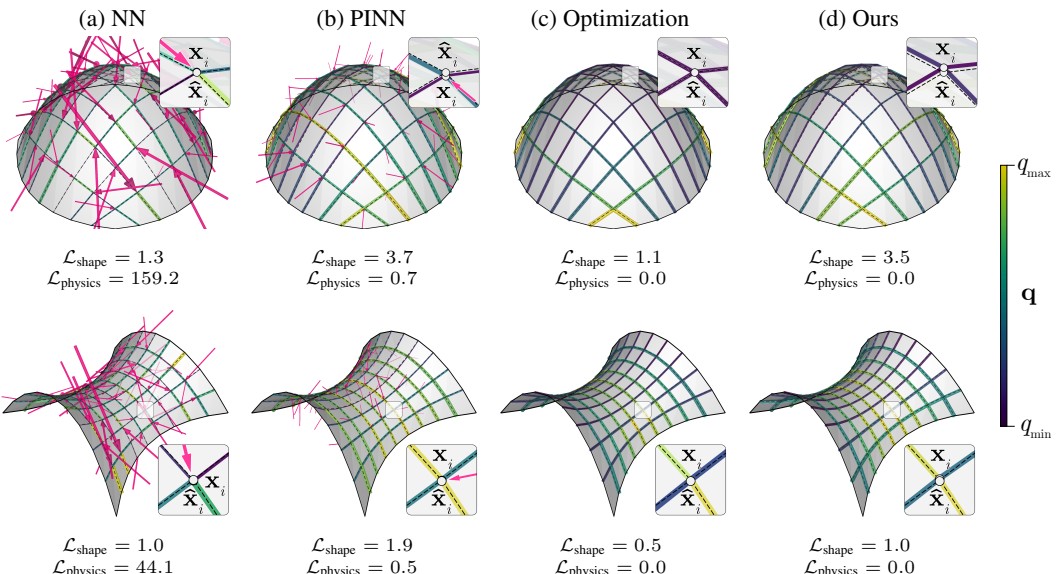

Figure 5: Shape matching for shell design. While the NN and PINN models approximate the targets, they cannot suppress the residual forces (pink arrows). The stiffnesses $\mathbf{q}$ and the shapes predicted by our model are similar to direct optimization's, indicating our model learns a good task representation.

Table 1: Model evaluation on the shell design task. We report the mean loss values and standard deviation on a test dataset of 100 target shapes, in addition to the inference wall time.

|  | NN | PINN | Optimization | Ours |
|---|---|---|---|---|
| $\mathcal{L}_{\text{shape}}\downarrow$ | $1.5\ \pm\ 0.4$ | $3.1\ \pm\ 1.2$ | $0.8\ \pm\ 1.2$ | $3.0\ \pm\ 2.0$ |
| $\mathcal{L}_{\text{physics}}\downarrow$ | $104.3\ \pm\ 48.6$ | $0.6\ \pm\ 0.3$ | $0.0\ \pm\ 0.0$ | $0.0\ \pm\ 0.0$ |
| Time [ms]$\downarrow$ | $0.3\ \pm\ 0.1$ | $0.3\ \pm\ 0.1$ | $5570.4\ \pm\ 1688.6$ | $0.6\ \pm\ 0.1$ |

## 4 EXPERIMENTS

We assess our model's ability to amortize shape approximation tasks for masonry shells and cable-net towers. These tasks represent a broad class of design problems in structural engineering.

### 4.1 MASONRY SHELLS

Our first experiment determines suitable stiffness values for unreinforced masonry shells. Masonry shells sustain external loads with span-to-thickness ratios as low as 1:50 despite being built from materials that are strong in compression and weak in other loading conditions (Block et al., 2017). Shapes that maximize internal compressive axial forces enable this efficient behavior.

An expressive family of shells can be constructed with surfaces parameterized by a Bezier patch. The shape of the patch is in turn described by the positions of a grid with $C$ control points (see Appendix C.1). For this task, we restrict the space of target shapes to a square grid of width $w = 10$ and $C = 16$ control points. In particular, we consider doubly symmetric shapes with $N = 100$ and $M = 180$. We employ limit state analysis to model masonry shells as pin-jointed bar systems (Maia Avelino et al., 2021), and apply a constant area load of $0.5$ per unit area, representing the self-weight of the shell. The nodes on the perimeter of the structures are fixed.

To solve this task, we are interested in finding bar stiffness values that yield shapes that fit the target geometries, and whose internal forces are compressive ($\mathbf{q} < \mathbf{0}$). We satisfy the compression-only requirement in the last layer of our encoder by setting $\mathbf{s} = -\mathbf{1}$. We use $\tau = 0$ and $p = 1$. Figure 4a illustrates the stochastic loss curves during training for our model and the two neural baselines (NN and PINN). The neural baselines achieve a low shape loss but are unable to converge w.r.t. the physics loss, unlike our model where this requirement is satisfied by construction.

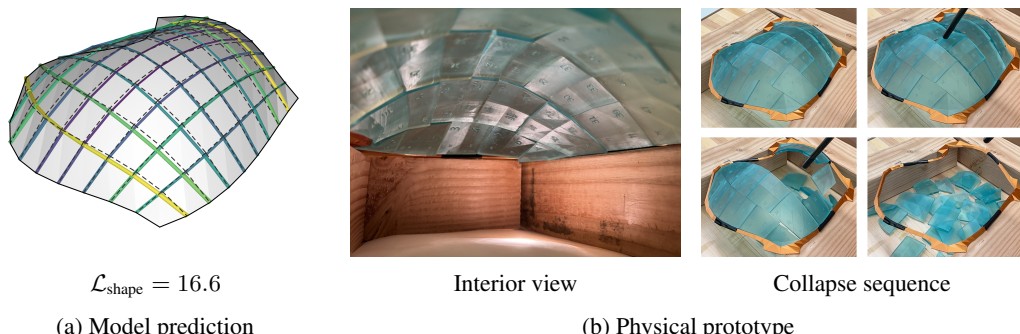

$\mathcal{L}_{\text{shape}} = 16.6$          Interior view          Collapse sequence

(a) Model prediction                (b) Physical prototype

Figure 6: (a) Our model accurately predicts asymmetric shapes despite being trained exclusively on doubly symmetric geometries. (b) We build a predicted shape as a tabletop prototype of a masonry shell. The bricks stand in equilibrium due to the appropriate shape; they are not mechanically attached as shown by the snapshots of the collapse sequence caused by an external perturbation.

The trends observed during training are consistent during inference (Table 1). Direct optimization achieves the lowest shape loss, but all the neural amortization models are faster than optimization. While the NN and the PINN generate close matches, their predictions are mechanically unfeasible because the residual forces do not vanish. The magnitude of the physics loss indicates that the structure is missing balancing forces to achieve equilibrium. For example, if we prescribe values for a masonry shell with a surface area of $10\text{m}^2$, thickness of 0.025 m, and material density of 2000 kg/m$^3$, then a residual force of 1 kN, representative of a unit value of $\mathcal{L}_{\text{physics}}$ in Table 1, would destabilize the structure with an acceleration of 2 m/s$^2$. Consequently, the NN and the PINN predictions are unstable, and constructing masonry shells guided by them can lead to cracks, joint separations, and ultimately to collapse. In contrast, our model and direct optimization fulfill the physics constraint from the outset. Although the loss of direct optimization is lower, our model still generates accurate predictions up to four orders of magnitude faster, and provides a similar speedup for predictions that incur an equivalent shape loss (see Appendix D.1). Figure 5 shows two representative targets alongside the shapes, stiffness values, and residuals predicted by our model and the baselines.

Next, we investigate the out-of-distribution generalization of our model and the PINN, an important property for building robust surrogates for physical design. To this end, we first generate a test dataset of 100 asymmetric Bezier surfaces and create target geometries by linearly interpolating between the existing doubly symmetric dataset and the new asymmetric dataset (Appendix C). We then evaluate the shape loss at increasing interpolation factors $\delta$, where $\delta = 0$ and $\delta = 1$ correspond to the doubly symmetric and asymmetric datasets, respectively. Note that both models have been trained solely on targets with double symmetry. A similar analysis for $\mathcal{L}_{\text{physics}}$ is provided in Appendix D.2. Our model exhibits superior out-of-distribution performance. As shown in Figure 4b, the shape loss of our model increases at half the rate of the PINN and with a lower variance. At $\delta = 1$, our model's shape loss is 2.5 times lower than that of the PINN. This performance gap is further illustrated in Figure 13 in Appendix D.2, where the PINN's prediction deviates from the asymmetric target by an order of magnitude more than in the doubly symmetric case, with a physics loss two orders of magnitude higher. In contrast, our model's shape loss increases by only one order of magnitude, while the physics loss remains at zero, evidencing that physics-in-the-loop models like ours offer enhanced generalization compared to physics-informed approaches.

To demonstrate the practical applicability of our model, Figure 14 in Appendix D.3 shows its integration into a 3D modeling environment, where it assists a designer in exploring various shell geometries. We further validate our model's generalization and transfer to physical applications by fabricating a tabletop masonry shell based on one of our model's predictions (Figure 6a). The asymmetric shell features a thickness-to-span ratio of 1:50 along its longest span. After tessellating the predicted shape, we manufacture and assemble the resulting bricks. The prototype remains stable and resists its weight without glue or fasteners, proving that the structure primarily resists gravity through internal compressive forces as required for masonry shells (Figure 6b). Moreover, in Appendix D.4, we solve this task at various discretizations and demonstrate that our model remains effective for real-time design at finer resolutions, all while maintaining prediction accuracy. These experiments confirm the feasibility of our model under real-world constraints.

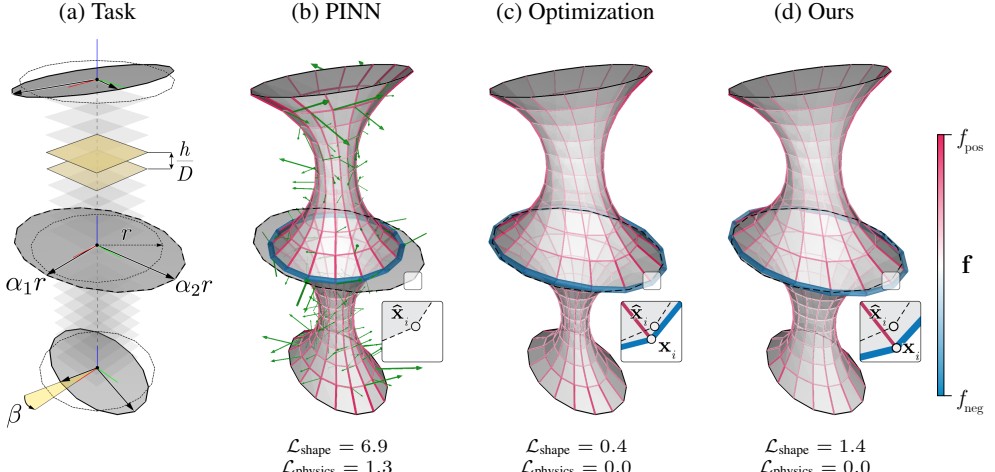

| (a) Task | (b) PINN | (c) Optimization | (d) Ours |
|---|---|---|---|

$\mathcal{L}_{\text{shape}} = 6.9$     $\mathcal{L}_{\text{shape}} = 0.4$     $\mathcal{L}_{\text{shape}} = 1.4$
$\mathcal{L}_{\text{physics}} = 1.3$     $\mathcal{L}_{\text{physics}} = 0.0$     $\mathcal{L}_{\text{physics}} = 0.0$

Figure 7: Predictions on cable-net structures. (a) Schematic depicting the design space. (b) - (d) Reconstructions of target shapes, showing internal tensile (red) and compressive (blue) forces $\mathbf{f}$. The PINN fails to match the target ellipse or ensure a net zero force balance, whereas our method closely approximates the target shape, similar to the solution obtained through direct optimization.

## 4.2 CABLE-NET TOWERS

Guided by the success of our initial experiment, we turn to a more complex problem: the design of cable-net towers. Such structures are lightweight, with low structural mass-to-volume ratios, and are often materialized as cooling or observation towers (Schober et al., 2014). To explore the design space of these towers, this task focuses on approximating the shape and orientation of a horizontal compression ring connected to two vertically spanning, tensile cable-nets.

Figure 7a gives an overview of the task setup. Every tower comprises $D = 21$ rings with 16 points each, spaced at equal $h/D$ intervals over a height of $h = 10$. The discretization of the structure is $N = 335$ and $M = 656$. We parametrize the geometry of the bottom, middle, and top rings with ellipses of radii $\alpha_1 r$ and $\alpha_2 r$, with $r = h/5$, and in-plane rotation angle $\beta$ (Appendix E.1). The nodes on the top and bottom rings are fixed. Besides matching the shape of the compression ring, we look for geometries where the tension rings are planar. The shape loss incorporates these requirements, measuring the squared $\ell_2$ norm (i.e., $p = 2$) between predictions and targets. We set the entries of $\mathbf{s}$ to $-1$ and $1$ to impose the force sign of the compression ring and the tension cables, respectively.

The mechanical behavior of cable-net towers poses modeling challenges because their structural members are either in tension or compression. This interplay results in an ill-conditioned problem, where interactions between elements of opposing force signs can lead to singular systems and instabilities near zero force levels (Cai et al., 2018). While ill-conditioning can be managed in the forward pass, it can hinder learning in the backward pass by producing poorly scaled gradients (Figure 8). To mitigate these numerical instabilities, we clip the global gradient norm to $0.01$ and shift the outputs of our last layer, enforcing a lower bound of $\tau = 1$ in the stiffness space. Additionally, we incorporate a regularization term $\mathcal{L}_{\text{reg}}$ into the training loss to encourage our model to learn balanced stiffness distributions:

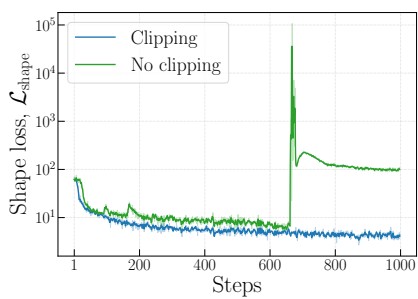

Figure 8: Training loss curve on the towers task, with and without clipping.

$$\mathcal{L}_{\text{reg}} = \text{Var}\left(\mathbf{q}_{\text{pos}}\right) + \text{Var}\left(\mathbf{q}_{\text{neg}}\right) \tag{7}$$

This regularizer measures the variance, $\text{Var}(\mathbf{q})$, of the bar stiffness values, for tensile $\mathbf{q}_{\text{pos}}$ and compressive $\mathbf{q}_{\text{neg}}$ components, over all the $B$ samples in a batch. We scale $\mathcal{L}_{\text{reg}}$ by a constant factor $\lambda$. We employ $\lambda = 10$ to train our model and the baselines.

Table 2: Model evaluation on the cable-net towers task. We report mean loss values and the standard deviation on a test dataset of 100 target shapes, in addition to the inference wall time.

| | NN | PINN | Optimization | | | Ours |
| --- | --- | --- | --- | --- | --- | --- |
| | | | Randomized | Expert | with Ours | |
| $\mathcal{L}_{\text{shape}}$ ↓ | 7.5 ± 3.4 | 7.4 ± 3.4 | 16.2 ± 28.2 | 0.3 ± 0.3 | 0.2 ± 0.3 | 0.6 ± 0.7 |
| $\mathcal{L}_{\text{physics}}$ ↓ | 45.8 ± 0.4 | 1.2 ± 0.1 | 0.0 ± 0.0 | 0.0 ± 0.0 | 0.0 ± 0.0 | 0.0 ± 0.0 |
| Time [ms] ↓ | 0.5 ± 0.1 | 0.5 ± 0.1 | 1690.4 ± 1169.9 | 1685.8 ± 663.9 | 1460.5 ± 823.4 | 4.6 ± 0.1 |

Figures 7b-7d show an example of the predicted cable-net tower shapes. In Appendix E.2, we show that our model predictions cover the task space satisfactorily, generating accurate and mechanically valid cable-net shapes whose compression ring radii change in size between $r/2$ and $3r/2$, and whose in-plane rotation vary within an angle range of $\pi/6$. Table 2 demonstrates that our model generates shapes that match the targets with a tighter fit than the NN and the PINN. These purely data-driven baselines make faster predictions because they do not run a physics simulator, but they are unable to learn the physics of this task because $\mathcal{L}_{\text{physics}}$ is nonzero, as we also identified in the shells task.

Furthermore, we compare the two approaches that guarantee physics: ours and direct optimization. The convergence of direct optimization depends on selecting appropriate initial stiffness values $\mathbf{q}$. In Figure 9, we therefore analyze the convergence rate of direct optimization with four different initializations. Random initial guesses of $\mathbf{q}$ converge to poor local optima as the shape loss is the highest among our experiments, underscoring the relevance of careful initialization to optimize structures with complex mechanical behavior. If $\mathbf{q}$ is handpicked by a human expert, optimization is more accurate than our model and the other baselines. Optimization with expert initialization achieves a shape loss equivalent to that of our model in a fifth of the convergence wall time reported in Table 2. Nevertheless, our trained model matches optimization in only one inference step that is three orders of magnitude faster and free of potentially expensive human intervention. These attributes make our model a more suitable approach to power automatic and interactive design tools.

Lastly, we investigate the effect of using the values of $\mathbf{q}$ predicted by our model and the PINN as inputs to direct optimization to refine the cable-net tower designs. This combination results in the most accurate matches, converging faster and consistently achieving a lower shape loss than direct optimization with expert initialized parameters (Figure 9). The PINN initialization converges more slowly and only matches the performance of our initialization towards the end of the convergence curve. In both cases, a neural model provides better initialization for optimization than the human expert, highlighting the potential of utilizing neural networks and standard optimization techniques in tandem to enhance mechanical design performance.

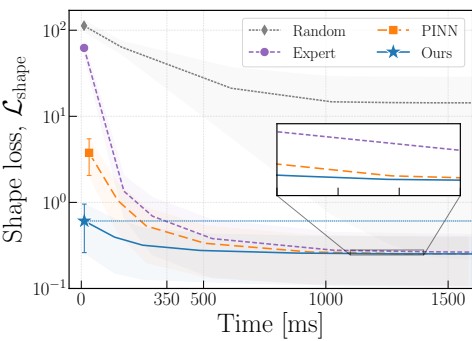

Figure 9: Convergence curves of direct optimization with four distinct initializations.

## 5 RELATED WORKS

**Differentiable mechanical simulators**  Machine learning and automatic differentiation have successfully been applied to obtain derivatives of complex forward mechanical models by either learning a differentiable surrogate with a neural network, or by implementing the analytical physics model in a differentiable programming environment. For neural surrogates, Xue et al. (2020) trained an autoencoder to simulate planar metamaterials, Zheng et al. (2020) developed a model to solve graphic statics problems on shells, and the work of Mai et al. (2024) focused on learning to simulate cable-nets under various boundary conditions. Differentiable mechanical simulators include the finite element method for solids (Xue et al., 2023) and isogeometric analysis of membranes (Oberbichler et al.,

2021). Specifically for form-finding, differentiable simulators exist for matrix structural analysis (Wu, 2023; Lee et al., 2024), and for combinatorial equilibrium modeling (Pastrana et al., 2023a). Our work leverages one such simulator and integrates it with a neural network to amortize inverse problems.

**Amortization models for mechanical design**   Neural surrogates that amortize inverse problems have garnered attention for their ability to approximate nonlinear relationships in mechanical systems between target properties and input parameters. At the centimeter scale, Bastek et al. (2022) addressed the inversion of the structure-property map in truss metamaterials, enabling the discovery of optimal configurations. Sun et al. (2021) synthesized path trajectories that compensate for fiber deformation during additive manufacturing. Oktay et al. (2023) utilized amortized models to generate actuation policies to deform cellular metamaterials to several target configurations. Focusing on architectural structures at the meter scale, Hoyer et al. (2019) proposed a neural basis for computing material distributions that minimize compliance for buildings in 2D, while Chang and Cheng (2020) amortized the design of the cross sections of the beams and columns of buildings in 3D for multiple load cases. Favilli et al. (2024) applied geometric deep learning and a differentiable simulator to gridshell structures for low strain energy shapes. Unlike their work, which trained a neural network for a single problem, we amortize over multiple inverse problems. Tam (2023) has amortized shape-matching problems with form-finding, similar to our work, but their approach relies entirely on neural networks, which must learn both the physics and the matching task simultaneously.

## 6   CONCLUSION

We present a physics-in-the-loop surrogate model that expedites the solution of shape-matching problems to design mechanically efficient geometry for architectural structures. By embedding prior physics knowledge into a neural network and training it end-to-end, our model learns representations that solve a family of inverse problems with precision while enforcing mechanical constraints by construction, which is where purely data-driven approaches fall short. While we expect that PINNs could eventually push the physics loss closer to zero in the limit of hyperparameter tuning and network scaling, the physics guarantees and superior generalization performance of our model make it a more robust approach to support design. Our model predicts mechanically sound geometries in milliseconds, with accuracy comparable to gradient-based optimization. The speed and reliability of our model enable real-time design exploration of long-span structures modeled as pin-jointed bar systems, such as masonry shells and cable-net towers.

### 6.1   LIMITATIONS AND FUTURE WORK

Although not a cure-all for mechanical design problems, our work evidences that domain expertise in structural engineering and machine learning is necessary to address numerical pathologies caused by incorporating physics into neural networks (Wang et al., 2021; 2022; Metz et al., 2022). In our case, the simulator can yield an ill-conditioned system that affects training in the presence of structures with complex force distributions. Overcoming these instabilities requires knowledge of the physics of the problem being modeled and the application of gradient stabilization techniques common in machine learning, such as gradient clipping.

Like past methods at this intersection, our model requires devising task-specific parameter spaces for learning (Allen et al., 2022). Consequently, a model trained for one task may not generalize to other tasks. Another limitation is that our model is currently restricted to one topology and has to be retrained when the discretization of a structure changes. In the future, we expect that applying methods such as graph networks (Battaglia et al., 2018) to our encoder will enable us to move beyond a single bar connectivity. We additionally note that the choice of bar stiffnesses for a given structure is not unique and it is potentially appealing to present to the designer a diversity of possible solutions by reformulating our model in a variational setting (Kingma and Welling, 2014; Salamanca et al., 2023). That is another exciting avenue for future research.

ACKNOWLEDGMENTS

The authors thank Deniz Oktay for the early discussions on this project. This work has been supported by the U.S. National Science Foundation under grant OAC-2118201.

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

## A    Mechanical simulator

We employ the force density method (FDM) (Schek, 1974) as our mechanical simulator. The FDM is a form-finding method that computes torsion- and bending-free shapes on pin-jointed bar systems with $N$ nodes and $M$ bars, by mapping bar stiffness values $\mathbf{q} \in \mathbb{R}^M$ to node positions $\mathbf{X} \in \mathbb{R}^{N \times 3}$, subject to boundary conditions $\mathbf{b} \in \mathbb{R}^L$. The stiffness $q_m$ or *force density* of a bar sets the ratio between its internal force $f_m$ and its length $l_m$ in the equilibrium configuration:

$$q_m = \frac{f_m}{l_m} \tag{8}$$

Since lengths are strictly positive, a negative $q_m$ indicates an internal compression force in the bar, and a positive $q_m$ denotes a tension force. In the FDM, a portion of nodes of size $N_s$ is fixed (i.e., anchors), and another of size $N_u$ is free to displace. The size of the partitions is arbitrary as long as $N = N_s + N_u$, but generally $N_u >> N_s$. For the geometry of a bar system to be in equilibrium, the residual force vector $\mathbf{r}_i$ at each free node $i$ must be zero. To satisfy this constraint, the FDM calculates the free node positions $\mathbf{X}_u \in \mathbb{R}^{N_u \times 3}$ by solving a linear system:

$$\mathbf{X}_u(\mathbf{q}) = \mathbf{K}(\mathbf{q})_u^{-1} \left[ \mathbf{P}_u - \mathbf{K}(\mathbf{q})_s \mathbf{X}_s \right] \tag{9}$$

where the submatrices $\mathbf{K}(\mathbf{q})_u \in \mathbb{R}^{N_u \times N_u}$ and $\mathbf{K}(\mathbf{q})_s \in \mathbb{R}^{N_u \times N_s}$ are created by the rows and the columns of the geometric stiffness matrix $\mathbf{K}(\mathbf{q}) \in \mathbb{R}^{N \times N}$ that correspond to the free and the fixed nodes in the structure, respectively. We solve this equation system with a direct, dense linear solver.

The FDM assembles $\mathbf{K}(\mathbf{q})$ based on the connectivity between bars and nodes (Vassart and Motro, 1999). The value of the $ij$-th entry in the stiffness matrix is defined as:

$$[\mathbf{K}(\mathbf{q})]_{ij} = \begin{cases} \sum_{m \in \mathcal{M}(i)} q_m & \text{if } i = j \\ -q_m & \text{if nodes } i \text{ and } j \text{ are connected by bar } m \\ 0 & \text{otherwise} \end{cases} \tag{10}$$

where $\mathcal{M}(i)$ denotes the bars connected to node $i$. The loads applied to the nodes $\mathbf{P} \in \mathbb{R}^{N \times 3}$ and the positions of the node anchors $\mathbf{X}_s \in \mathbb{R}^{N_s \times 3}$ are the boundary conditions of the problem, so they are inputs to the simulator. To create the boundary conditions vector $\mathbf{b}$, we concatenate $\mathbf{P}$ and $\mathbf{X}_s$, and then flatten the resulting matrix into a vector. Once we know $\mathbf{X}_u$, we concatenate it with $\mathbf{X}_s$ to obtain all the node positions $\mathbf{X}$ that characterize a bar structure in equilibrium.

## B    Implementation details

We implement our work in JAX (Bradbury et al., 2018) and Equinox (Kidger and Garcia, 2021). We use JAX FDM (Pastrana et al., 2023b) as our differentiable simulator and Optax (DeepMind et al., 2020) for derivatives processing. For our model and the neural baselines (NN and PINN), we train a separate model instance per task to minimize the loss expectation over a batch of shapes of size $B$. For optimization, we directly minimize the loss for every shape in the batch, one shape at a time.

Training and inference for all models are executed on a CPU, on a Macbook Pro laptop with an M2 chip. This choice of hardware is motivated by the type of devices structural engineers have access to and also reveals the potential for enhanced model performance on more powerful hardware, like GPUs, in the future. We train the three models using Adam (Kingma and Ba, 2017) with default parameters, except for the learning rate, for 10,000 steps. The training hyperparameters per model are tuned to maximize predictive performance on the test dataset via a random search over three seeds.

We employ SLSQP (Kraft, 1994) and L-BFGS-B (Zhu et al., 1997), two deterministic gradient-based optimizers, as the direct optimization baselines. We use their implementations in JAXOPT (Blondel et al., 2022) with default settings. To make a fair comparison, we also perform all the direct optimization experiments on the same laptop and CPU. We run these gradient-based optimizers for at most 5,000 steps and with a convergence tolerance of $1 \times 10^{-6}$.

In direct optimization, we impose box constraints on the bar stiffness values $\mathbf{q}$ (the optimization variables) to prescribe an explicit lower bound on the force signs and magnitudes. This mimics the effect of the last layer of our encoder via $\tau$ (see Section 2.3). We use the same values of $\tau$ that we report in Section 4 as the lower box constraints. Additionally, we set an upper box constraint on each

Table 3: Neural architectures. We show the composition of the two types of autoencoders we use in Section 4: a fully neural model employed for the NN and PINN models, and ours with the differentiable simulator as an analytical decoder. In this table, Linear(in_size, out_size) denotes a linear layer with learnable weights and biases followed by a nonlinearity; whereas Concat(in_size, out_size) denotes the concatenation of two vectors.

| Task | Model | Encoder | Decoder | # Parameters ↓ | Train time (s) ↓ |
|------|-------|---------|---------|----------------|------------------|
| Shells | NN, PINN | [Linear$(3N, H)$; $2\times$ Linear$(H, H)$; Linear$(H, M)$] | [Linear$(M + L, H)$; $2\times$ Linear$(H, H)$; Linear$(H, 3N_u)$; Concat$(3N_u + 3N_s, 3N)$] | 535,412 | 85 |
| | Ours | [Linear$(3N, H)$; $2\times$Linear$(H, H)$; Linear$(H, M)$] | Simulator$(M + L, 3N)$ | 254,900 | 140 |
| Towers | NN, PINN | [Linear$(3R, H)$; $4\times$ Linear$(H, H)$; Linear$(H, M)$] | [Linear$(M + L, H)$; $4\times$ Linear$(H, H)$; Linear$(H, 3N_u)$; Concat$(3N_u + 3N_s, 3N)$] | 1,245,216 | 136 |
| | Ours | [Linear$(3R, H)$; $4\times$Linear$(H, H)$; Linear$(H, M)$] | Simulator$(M + L, 3N)$ | 468,880 | 810 |

bar stiffness $q$ so that $|q| \leq 20$, where $|\cdot|$ denotes the absolute value. Note that we do not enforce an upper bound on $\mathbf{q}$ in any of the neural models during training or inference.

Table 3 summarizes the configuration of the neural models studied herein. We utilize MLPs for the learnable components of the three models. The MLPs are a sequence of linear layers followed by a nonlinear activation function. The weights and biases per layer are initialized at random from a uniform distribution in the interval $[-1/\sqrt{\rho}, 1/\sqrt{\rho})$, where $\rho$ is equal to the layer's input size. We use an ELU (Clevert et al., 2016) as the activation function in every MLP layer except in the last layer. The MLP encoders utilize Softplus as the last nonlinearity to satisfy the strictly positive requirement of our simulator (Section 2.3), and the MLP decoders use the identity function. Our model is more compact than the neural baselines because it has fewer than half the number of trainable parameters. However, our model takes longer to train due to the computational cost of the mechanical simulator.

The MLP decoders in the NN and PINN baselines take as input the stiffness values $\mathbf{q}$ predicted by the encoder, in addition to a vector with the boundary conditions $\mathbf{b}$ so that the learnable decoder sees as much of the information the mechanical simulator has access to. This is different from, for example, conventional PINN approaches where the boundary conditions should be learned from a loss function, in addition to the problem physics (Raissi et al., 2019a; Haghighat et al., 2021); but akin to the approach proposed in other recent works where boundary conditions are treated as either hard optimization constraints or imposed as network inputs (Lu et al., 2021; Bastek and Kochmann, 2023; Mai et al., 2024). Here, we simplify the decoder's goal by imposing the boundary conditions by construction, like in the latter case. This is why the boundaries of the shape targets are matched exactly in all of our examples. For simplicity, $\mathbf{b}$ only comprises the positions of the $N_s$ fixed nodes and the vertical component of the loads applied to the $N$ nodes (i.e., the horizontal components are zero) per target shape. Therefore, $\mathbf{b}$ has size $L = 3N_s + N$ for all the neural decoders.

The last layer in the MLP decoders outputs a vector of size $3N_u$, representing the position of the free nodes, $\mathbf{X}_u$. This choice is motivated by the fact that the mechanical simulator solves the form-finding problem exclusively at the free nodes of the bar system with Equation 9, and subsequently concatenates the known positions of the support nodes, $\mathbf{X}_s$, to assemble the full shape matrix $\mathbf{X}$. We replicate this process in the MLP decoder, ensuring that it predicts the same amount of information as the simulator. We describe task-specific implementation details next.

## B.1 Masonry shells

For the shells task, we train MLPs with 2 hidden layers with $H = 256$ units each. The size of the input and output layers varies as per Table 3. The batch size is $B = 64$. The learning rate is $3 \times 10^{-5}$ for the fully neural baselines (NN and PINN), and $5 \times 10^{-5}$ for our model. Our model takes 2 minutes and 20 seconds to train. The fully neural alternatives, in contrast, take 1 minute and 25 seconds. We utilize SLSQP (Kraft, 1994) as the direct optimization baseline, sampling random values of $\mathbf{q}$ from a uniform distribution bounded by the box constraints as initial guesses.

We train the PINN baseline to minimize a weighted combination of the shape loss $\mathcal{L}_{\text{shape}}$ (Equation 2) and the physics loss $\mathcal{L}_{\text{physics}}$ (Equation 6):

$$\mathcal{L} = \mathcal{L}_{\text{shape}} + \kappa \mathcal{L}_{\text{physics}} \tag{11}$$

Different values of $\kappa$ impact predictive performance on this bipartite task at inference time (i.e., simultaneously matching target shapes and reducing the residual forces). To strengthen the PINN baseline, we tune $\kappa$ in five increments, from $\kappa = 10^{-2}$ to $\kappa = 10^2$, and train a separate instance of the model for each. We then pick the PINN that achieves the lowest unweighted loss on the test dataset (i.e., we set $\kappa = 10^0$ during inference to evaluate performance). Table 4 shows our results. We find that employing $\kappa = 10^1$ during training produces the best-performing PINN baseline, and we use this PINN for comparison with our model and the other baselines in Section 4.

Table 4: PINN evaluation on the masonry shells task for five different coefficients $\kappa$ applied to $\mathcal{L}_{\text{physics}}$ during training. The table reports the unweighted loss values generated by the trained PINN model predictions at inference time on the test dataset.

| $\kappa$ | $10^{-2}$ | $10^{-1}$ | $10^0$ | $10^1$ | $10^2$ |
|---|---|---|---|---|---|
| $\mathcal{L}_{\text{shape}}$ ↓ | $1.9 \pm 0.4$ | $1.5 \pm 0.3$ | $1.4 \pm 0.3$ | $\mathbf{3.1 \pm 1.2}$ | $92.0 \pm 40.5$ |
| $\mathcal{L}_{\text{physics}}$ ↓ | $35.3 \pm 16.7$ | $6.3 \pm 2.8$ | $2.3 \pm 1.0$ | $\mathbf{0.6 \pm 0.3}$ | $0.3 \pm 0.1$ |

## B.2 Cable-net towers

In the cable-net towers task, we train MLPs with 4 hidden layers of size $H = 256$. In all the models, we set the batch size to $B = 16$, and reduce the size of the encoder input from $3N$ to $3R$, where $N$ is the total number of nodes in the structure, and $R = 48$ corresponds to the number of nodes on the three rings (bottom, middle, and top) that parametrize this task. This choice reduces the total number of trainable parameters in the encoders and makes them less computationally intensive.

The shape approximation task for the towers is underspecified because it only prescribes target height values for the tension rings of the cable-nets, rather than specific target 3D coordinates. Therefore, we mask (i.e., multiply by zero) the predicted $x$ and $y$ coordinates of the nodes of the tension rings before evaluating the shape loss.

In this task, we train our model in two stages, reducing the optimizer's learning rate from one stage to the next. In the first stage, we optimize for 5,000 steps with a learning rate of $1 \times 10^{-3}$. In the second stage, we fine-tune our model for another 5,000 steps with a lower learning rate of $1 \times 10^{-4}$. The total training time is 13.5 minutes. We train the MLPs of the NN and the PINN with a learning rate of $1 \times 10^{-3}$ over 10,000 steps. The training time of both models is 2 minutes and 16 seconds. The global gradient clip value is 0.01 for all the neural models.

The loss function we utilize to train the PINN baseline contemplates the shape and the physics losses, in addition to the regularization term (Equation 7) scaled by $\lambda = 10$:

$$\mathcal{L} = \mathcal{L}_{\text{shape}} + \kappa \mathcal{L}_{\text{physics}} + \lambda \mathcal{L}_{\text{reg}} \tag{12}$$

Like in the shells task, we vary the value of the weight coefficient $\kappa$ between $\kappa = 10^{-2}$ and $\kappa = 10^2$. As shown in Table 5, training the PINN with $\kappa = 10^0$ yields the best performance on the test dataset. We use this PINN variant for comparison with our model and the other baselines.

We employ L-BFGS-B (Zhu et al., 1997), a quasi-Newton optimizer, as the direct optimization baseline in this task. We initialize $\mathbf{q}$ for direct optimization with four different approaches: randomized,

Table 5: PINN model evaluation on the cable-net tower task for different coefficients $\kappa$ applied to $\mathcal{L}_{\text{physics}}$ during training. The table reports unweighted loss values generated by the trained PINN predictions on the test dataset at inference time.

| $\kappa$ | $10^{-2}$ | $10^{-1}$ | $10^{0}$ | $10^{1}$ | $10^{2}$ |
|---|---|---|---|---|---|
| $\mathcal{L}_{\text{shape}}\downarrow$ | $42.8 \pm 17.8$ | $9.5 \pm 5.5$ | $\mathbf{7.4 \pm 3.4}$ | $4.2 \pm 2.6$ | $5.1 \pm 3.5$ |
| $\mathcal{L}_{\text{physics}}\downarrow$ | $1.1 \pm 0.1$ | $2.1 \pm 0.0$ | $\mathbf{1.2 \pm 0.1}$ | $3.8 \pm 0.7$ | $4.3 \pm 0.6$ |

expert, with the trained PINN, and with our trained model. The randomized initialization samples $\mathbf{q}$ from a uniform distribution bounded by the box constraints. The expert initialization sets $\mathbf{q} = \mathbf{1}$. Both initialization approaches set the sign of the bar forces as specified by the task: negative signs for compression and positive signs for tension.

## C  DATA GENERATION ON BEZIER SURFACES

A Bezier patch $\mathcal{B}$ maps a matrix $\mathbf{C} \in \mathbb{R}^{C \times 3}$ of control points to a smooth surface in $\mathbb{R}^3$, $\mathcal{B} : \mathbb{R}^{C \times 3} \to \mathcal{S}(u, v)$, parameterized by local coordinates $u, v \in [0, 1]$. This parametrization offers clear control over architectural design intent as it enables the exploration of a wide array of smooth geometries by simply changing the positions of a coarse control grid.

### C.1  DATA GENERATION FOR SHELLS

A summary of the data generation process is given in Figure 10. To generate a family of shapes for the shells task in Section 4, we focus on doubly symmetric shapes produced by a square grid $w = 10$ units wide centered on the origin. The grid contains $C = 16$ control points arranged in a $4 \times 4$ layout.

We then follow three main steps. First, we vary the 3D coordinates of the control points $\mathbf{c}_1$ to $\mathbf{c}_3$ on a quarter of the control grid. The position of $\mathbf{c}_4$ is static. This construction ensures double symmetry in the generated data. To vary the position of every movable control point, we first sample a translation vector $\mathbf{t}$ at random from a uniform distribution in the interval $[\mathbf{t}_{\min}, \mathbf{t}_{\max})$; and then we add it to its reference position $\mathbf{c}_r$, such that $\mathbf{c} = \mathbf{c}_r + \mathbf{t}$. We detail the reference positions and the translation intervals for each control point in Table 6.

Next, we mirror the four control points on the $xz$ and the $yz$ Cartesian planes to obtain the position of the remaining twelve control points on the grid (see the callout in Figure 10). The bounding box of the resulting design space has dimensions $2w \times 2w \times w$. In the third and last step, we evaluate $N$

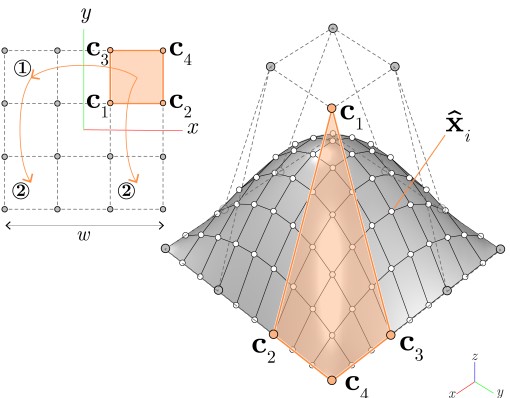

Figure 10: The data generation process of doubly symmetric shapes for the shells task consists of, first, creating variations of the positions of the points $\mathbf{c}_1$ to $\mathbf{c}_3$ in a corner of the control grid of a Bezier surface, and, then, mirroring twice. We create target points $\hat{\mathbf{x}}_i$ by evaluating the Bezier at $N$ different locations on the surface.

Table 6: Generation parameters to sample random variations of the 3D coordinates of the control points $c_1$ to $c_4$ on a quarter of the control grid of a Bezier surface.

| Control point | Reference position, $c_r$ | Lower bound, $t_{min}$ | Upper bound, $t_{max}$ |
|:---:|:---:|:---:|:---:|
| $c_1$ | $[w/6, w/6, 0]$ | $[0, 0, w/10]$ | $[0, 0, w]$ |
| $c_2$ | $[w/2, w/6, 0]$ | $[-w/2, 0, 0]$ | $[w/2, 0, w/2]$ |
| $c_3$ | $[w/6, w/2, 0]$ | $[0, -w/2, 0]$ | $[0, w/2, 0]$ |
| $c_4$ | $[w/2, w/2, 0]$ | — | — |

coordinate values $u$ and $v$, equally spaced in a $\sqrt{N} \times \sqrt{N}$ grid, to generate an equal number of points on the surface. The evaluated points represent the position $\hat{X} \in \mathbb{R}^{N \times 3}$ of the nodes of the structure we employ as targets to train our model and the baselines. The coordinates of a point $\hat{x} \in \mathbb{R}^3$ on the Bezier surface are a function of a $(u, v)$ coordinates pair:

$$\hat{x}(u, v) = \sum_{e=1}^{4} \sum_{g=1}^{4} \gamma_e(u) \, \gamma_g(v) \, c_{eg} \tag{13}$$

where $\gamma$ denotes a Bernstein polynomial of degree 3, and $c_{eg}$ indicates the position of the Bezier's control points, indexed on the $4 \times 4$ grid.

### C.2 INTERPOLATION OF BEZIER SURFACES

We utilize linear interpolation to blend between doubly symmetric and asymmetric Bezier surfaces. Since the targets $\hat{X}$ are a function of the control points matrix $C$, we directly interpolate between the control points matrix $C_{sym}$ of a doubly symmetric surface and that of an asymmetric surface $C_{asym}$ to create one design:

$$C_{interp} = (1 - \delta) C_{sym} + \delta C_{asym} \tag{14}$$

where $\delta \in [0, 1]$ is a scalar interpolation factor and $C_{interp}$ is the interpolated control points matrix. We vary all the control points in $C_{asym}$ by sampling random translation vectors $t$ from a uniform distribution with the same bounds as in the doubly symmetric case (Table 6). We finally generate $\hat{X}$ from $C_{interp}$ with Equation 13. Figure 13a provides an example of the shapes resulting from interpolating two surfaces.

## D ADDITIONAL STUDIES FOR SHELL DESIGN

### D.1 COMPARISON WITH DIRECT OPTIMIZATION

Both our model and direct optimization satisfy the physical constraints a priori, but they generate predictions at drastically different speeds. Since optimization is an iterative approach, we compare the evolution of the shape loss over time to that of our method, which predicts the bar stiffness values of a structure in one step. Optimization reaches the best predictive performance among the baselines, but only after convergence, which takes over 5000 milliseconds, as shown in Figure 11. Even though optimization matches the shape loss of our model in 2000 milliseconds (35% of the convergence wall time in Table 1), our approach offers a substantial speedup as it generates designs of equivalent accuracy in a millisecond.

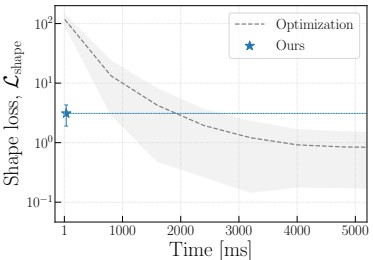

Figure 11: Shape loss evolution of direct optimization for shell design.

## D.2 GENERALIZATION ON THE PHYSICS LOSS

In Section 4.1, we discussed the generalization capacity of our model and a PINN—the approaches that are aware of the inverse problem physics—by observing the rate of change of the shape loss $\mathcal{L}_{\text{shape}}$ as we morph the targets between doubly symmetric ($\delta = 0$) and asymmetric ($\delta = 1$) datasets.

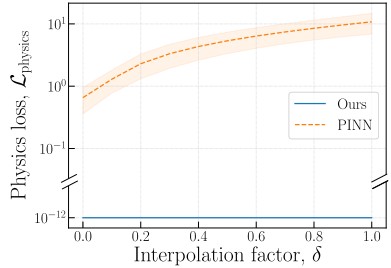

We now quantify the impact of perturbing the test data distribution on the models' capacity to preserve the problem's physics. Figure 12 depicts the evolution of the physics loss $\mathcal{L}_{\text{physics}}$ in logarithmic scale. The performance of the PINN deteriorates rapidly as the distribution shifts from the doubly symmetric to the asymmetric dataset, with the loss increasing by an order of magnitude.

Figure 12: Physics loss changes as the shapes vary between $\delta = 0$ and $\delta = 1$.

This finding underscores that the in-distribution performance w.r.t. physics of the PINN does not necessarily lead to out-of-distribution generalization. In contrast, our model enforces the physics by design, so $\mathcal{L}_{\text{physics}}$ remains constant and effectively zero because of our mechanical decoder, regardless of the geometry of the targets. Figure 13 illustrates this trend with an example.

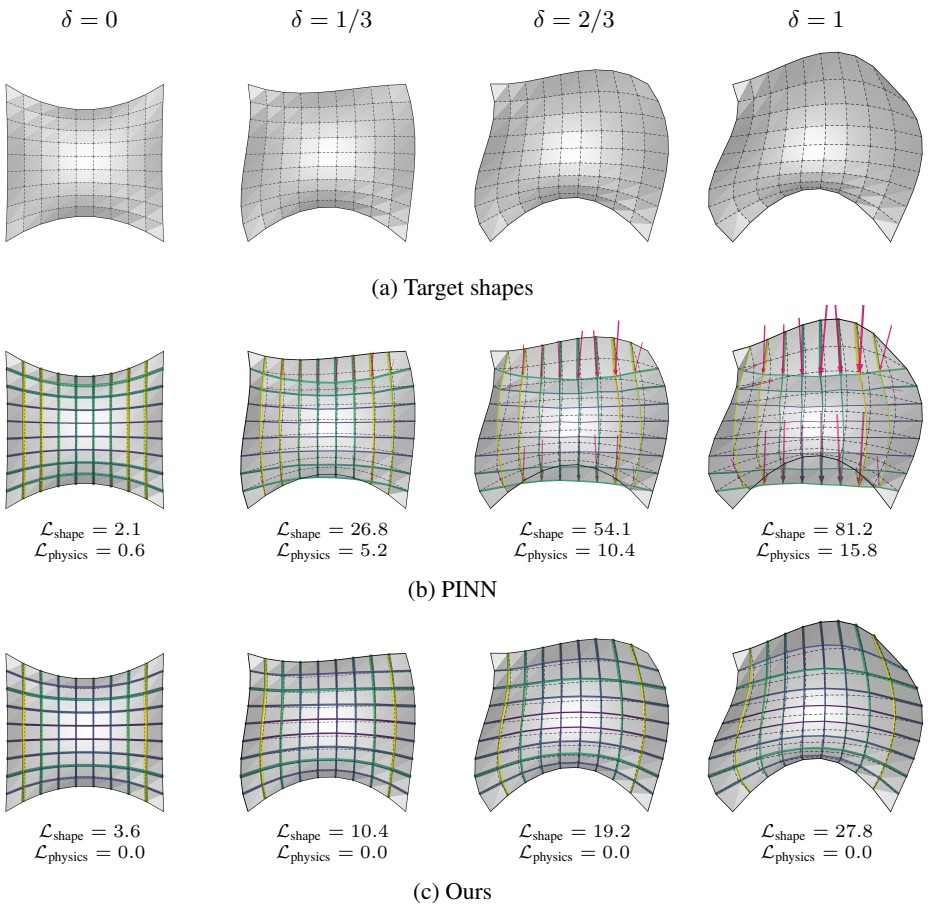

Figure 13: Top view of predicted shapes for shell design. (a) The targets are interpolated with different interpolation factors $\delta$. (b) The accuracy of the PINN decays quickly w.r.t. $\mathcal{L}_{\text{shape}}$ and $\mathcal{L}_{\text{physics}}$ as $\delta$ increases. (c) Our model's shape accuracy deteriorates, but at a lower rate, and the problem's physics are fulfilled by construction since $\mathcal{L}_{\text{physics}} = 0$ within numerical precision.

## D.3 REAL-TIME DESIGN IN CAD SOFTWARE

We deploy our trained model in the 3D modeling software Rhino3D (Robert McNeel & Associates, 2024) as illustrated in Figure 14. Rhino3D supports traditional computer-aided design (CAD) work-flows and Grasshopper (Figure 14, top left), its visual programming extension, allows the creation of new software features via custom Python scripts. We load our model as a Grasshopper plugin via Python and test it to support the real-time exploration of shapes for shell structures.

We describe a design exploration session next. A designer draws a Bezier surface in Rhino3D and imports it into Grasshopper. Then, as the designer moves the control points of the Bezier, the geometry updates automatically. Our model generates new compression-only shapes in response to the changes (see the bar systems rendered in Figure 14). Note that as we detail in Section 4, we trained our model on doubly symmetric shapes, but the adequate generalization it exhibits to asymmetric geometries makes it possible to support the designer during their exploratory session.

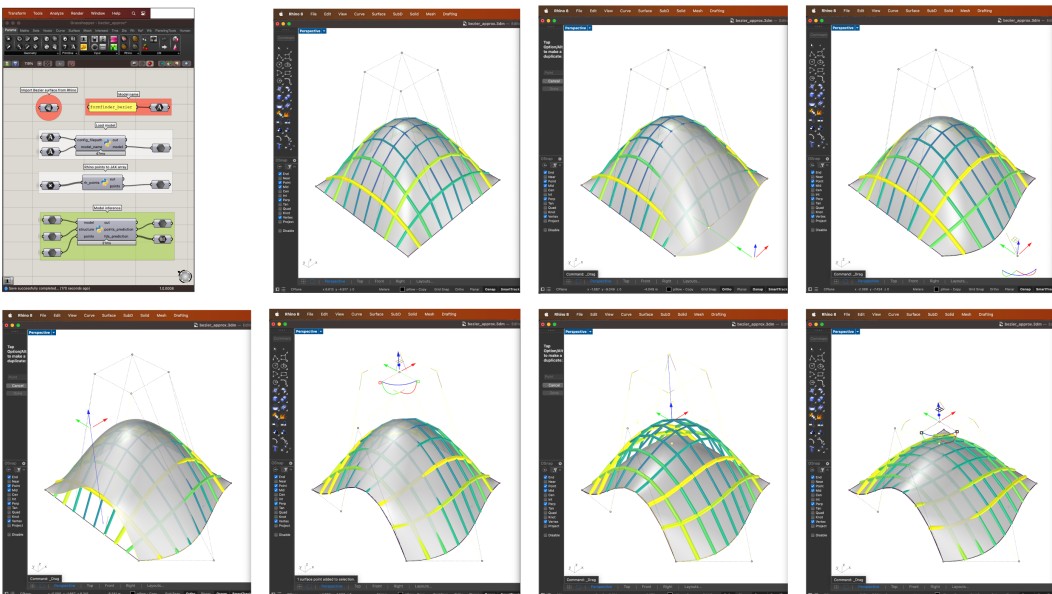

Figure 14: A demonstration of our trained model providing predictions of compression-only shapes in real time. The first screenshot shows the environment running our custom code. The screenshot in the second column and first row shows the initial input Bezier surface with control points. Every subsequent screenshot shows the designer moving Bezier control points and our model reacting to the designer's intent by approximating the shape with one that is compression-only.

## D.4 TRAINING COST AMORTIZATION

We measure trade-offs between our model and direct optimization in terms of training versus test inference time in problems where $N > 500$ and $M > 1000$. $N$ and $M$ are the numbers of nodes and bars in a structure, respectively. These experiments also reflect the overhead of the mechanical simulator during training, as increasing the problem size also enlarges $\mathbf{K}$, the bottleneck in Equation 4. We keep the architecture of our MLP encoder, except for the input dimension of the first layer, and the output dimension in the last layer, matching the new values of $N$ and $M$. The settings for training and direct optimization repeat the configurations described in Appendix B. The size of the test dataset is $B = 100$, except in optimization for $M = 1012$ where $B = 10$ due to hardware limitations.

Table 7 demonstrates that our model remains a feasible approach to amortize inverse problems on denser structures. While training and inference times increase with problem size, the expected inference time at the finest discretization is just ten milliseconds, making it still suitable for real-time design exploration. Remarkably, the shape loss $\mathcal{L}_{\text{shape}}$ normalized by $N$ changes marginally as the problem size grows, revealing the ability of our model to generate adequate fits to the target shapes across several discretizations despite the minimal changes in the encoder architecture (Figure 15).

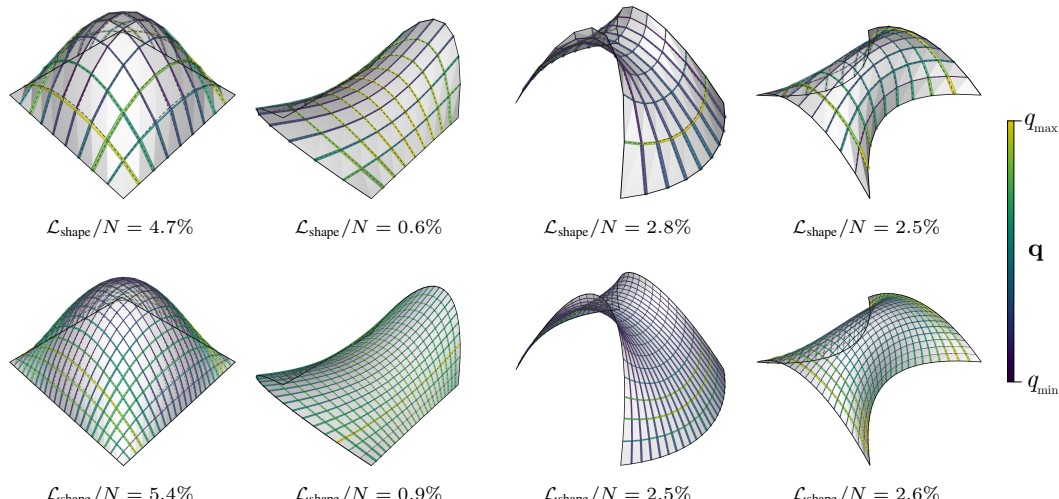

Figure 15: Our model makes accurate predictions across multiple discretizations. Top row: $N = 100$. Bottom row: $N = 529$. The predictions satisfy the problem physics and are made in milliseconds.

Table 7: Comparison between our model and direct optimization for varying problem sizes in the shells task. We report the mean loss values and standard deviations normalized by $N$ on a test dataset, and the inference wall time. The column # Opt lists the number of optimization problems required to match or exceed the training time of our model, which corresponds to the number of inference calls needed to justify our model's training cost.

| $N$ | $M$ | Optimization | | Ours | | | |
|---|---|---|---|---|---|---|---|
| | | $\mathcal{L}_{\text{shape}}/N$ [%] $\downarrow$ | Time [s] $\downarrow$ | $\mathcal{L}_{\text{shape}}/N$ [%] $\downarrow$ | Test time [ms] $\downarrow$ | Train time [s] $\downarrow$ | # Opt $\downarrow$ |
| 100 | 180 | $0.8 \pm 1.2$ | $5.6 \pm 1.7$ | $3.0 \pm 2.0$ | $0.6 \pm 0.1$ | 140 | 25 |
| 256 | 480 | $1.0 \pm 1.5$ | $287.7 \pm 115.8$ | $3.6 \pm 2.3$ | $3.5 \pm 0.1$ | 1928 | 7 |
| 529 | 1012 | $0.8 \pm 0.9$ | $5483.1 \pm 1946.0$ | $3.5 \pm 2.4$ | $10.2 \pm 0.3$ | 6405 | 2 |

The amortization cost of training our model decreases relative to direct optimization as the problem size goes up (Table 7). Compared to the reference case in Section 4.1, the time needed to match one shape with optimization grows by two orders of magnitude for two times more bars, and by three orders of magnitude for five times as many bars. As a result, the number of shape matches needed to justify training our model is lowered by a factor of three every time the number of bars doubles.

# E  SHAPE EXPLORATION FOR CABLE-NET TOWER DESIGN

## E.1  DATA GENERATION

To generate data for the cable-net tower task described in Section 4.2, we parametrize the geometry of the bottom, middle, and top rings with ellipses of radii $\alpha_1 r$ and $\alpha_2 r$, and rotation angle $\beta$ on the plane. Once we set the reference radius $r$ to $h/5$, we create the training and testing datasets by sampling the scale factors $\alpha_i \in [1/2, 3/2)$ and the angles $\beta \in [-\pi/12, \pi/12)$ from a uniform probability distribution. See Figure 7a for a graphical description.

## E.2  SHAPE EXPLORATION

We carry out two experiments that modify the size and the rotation of the middle compression ring in a cable-net tower in order to illustrate the range of our trained model's predictions. In both experiments, we fix $\alpha_1 = \alpha_2 = 3/4$ at the top and bottom rings to form circular anchors. In the first experiment, shown in Figure 16, we modify the value of $\alpha_1$ and $\alpha_2$ of the middle compressive ring from $\alpha_1 = 1/2$

to $\alpha_1 = 3/2$, and from $\alpha_2 = 1/2$ to $\alpha_2 = 3/2$, in five steps. In the second experiment, shown in Figure 17, the compression ring is elliptical from the start ($\alpha_1 = 1/2$, $\alpha_2 = 3/2$) and we rotate it on the horizontal plane by an angle $\beta$ between $-\pi/12$ and $\pi/12$, also in five steps. The predictions in both experiments match the target geometry. Note that the distribution of the bar forces $\mathbf{f}$ changes per prediction to maintain force equilibrium while matching the target shape of the compression ring.

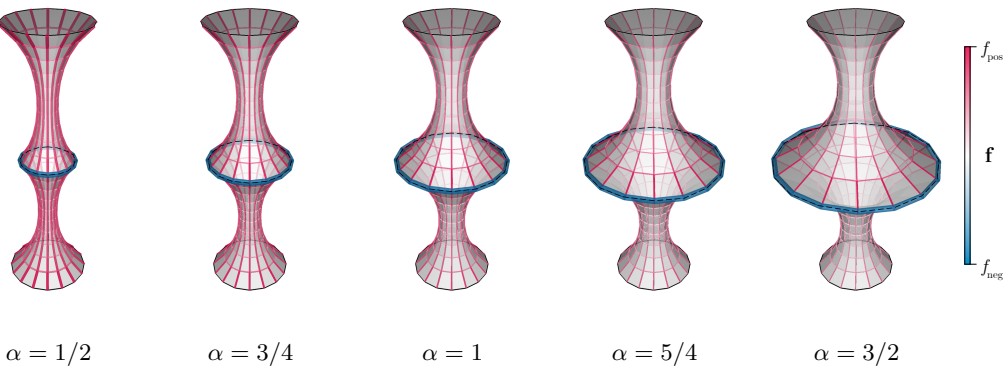

| $\alpha = 1/2$ | $\alpha = 3/4$ | $\alpha = 1$ | $\alpha = 5/4$ | $\alpha = 3/2$ |

Figure 16: Our model predictions for the cable-net tower design task, where we gradually increase the radius scale factor $\alpha$ of the target circle of the compression ring from $\alpha = 1/2$ to $\alpha = 3/2$.

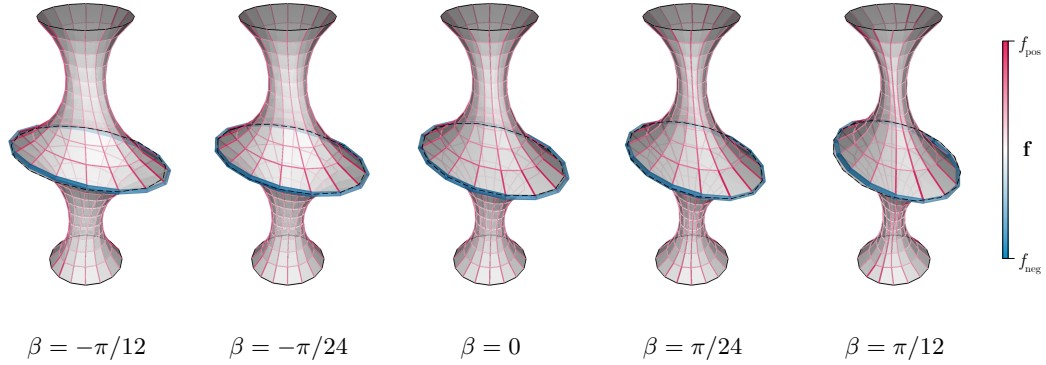

| $\beta = -\pi/12$ | $\beta = -\pi/24$ | $\beta = 0$ | $\beta = \pi/24$ | $\beta = \pi/12$ |

Figure 17: Our model produces physically valid predictions for the design of a cable-net tower with an elliptical compression ring, as we rotate the ring by an angle $\beta$ between $-\pi/12$ and $\pi/12$.

