# OpenReview forum: "Real-time design of architectural structures with differentiable mechanics and neural networks"
_ICLR.cc/2025/Conference — ICLR 2025 Poster_

### Official Review · Reviewer_H1FF · 2024-10-31

**Soundness:** 3
**Presentation:** 3
**Contribution:** 3
**Rating:** 8
**Confidence:** 3

**Summary:**

The authors propose an AI-driven design algorithm for architectural structure that can generate mechanically feasible designs for bar systems according to a target geometry. Previous approaches such as pure neural networks cannot guarantee mechanical design requirements. Traditional direct optimization methods can satisfy mechanical criteria, however are inefficient. The authors propose a hybrid scheme, combining neural networks and mechanical simulators to achieve computational efficiency and satisfaction of mechanical constraints. The authors approach consists of a encoder-decoder neural network, where the decoder is replaced by a differentiable mechanical simulator. The encoder structure provides efficiency, while the decoder ensures the design satisfies mechanical constraints. Experimental results show that the authors method achieves satisfactory performance in terms of design and computational efficiency.

**Strengths:**

The paper presentation is very good, the motivation is clear, and the method is justified well. Experimental results are extensive. The authors also provide a real life sample designed from their algorithm, which the readers will find very interesting.

**Weaknesses:**

The authors could do some revision of the figure. In figure 1, it seems that 3 figures are simultaneously provided to the encoder network, although the text description indicates only one shape is provided. Please take care to avoid such confusion from readers. Also describe the dimension of the matrix K. In figure 3, it is hard to see how the x vectors are being updated. Zooming in and adding some texts will be better for understanding.

**Questions:**

See weakness section

---

> ### Author Response · Authors · 2024-11-22
>
> 1. **In Figure 1, it seems that 3 figures are simultaneously provided to the encode, but the caption indicates only one shape is provided. Please take care to avoid confusion.**
>
> Thank you for noting this important detail. We have updated Figure 1 to match the caption, and to indicate clearly that one shape is fed to the encoder.
>
> 2. **Describe the dimension of the matrix K.**
>
> Certainly! We have explicitly indicated the dimension of matrix $\mathbf{K}$ in Section 2.1 and in Appendix D, where we detail the math behind the mechanical simulator.
>
> 3. **In Figure 3, it is hard to see how the x vectors are being updated.**
>
> We have retouched Figure 3 to improve legibility. We increased the size of the zoomed-in window (i.e., the callouts) in each step of the optimization, decreased the opacity of the background images in the callouts, and thickened the orange arrows that indicate the updates of $\mathbf{x}_i$.

---

### Official Review · Reviewer_jNkr · 2024-10-31

**Soundness:** 3
**Presentation:** 3
**Contribution:** 2
**Rating:** 6
**Confidence:** 2

**Summary:**

This paper combines neural networks with a differentiable mechanics simulator to develop a model that accelerates the solution of shape approximation problems for architectural structures modeled as bar systems. The proposed approach achieves better accuracy and generalization than the neural network approach and gets comparable accuracy to direct optimization. The two design domains are masonry shells and cable-net towers.

**Strengths:**

* The presentation of the approach is nicely structured, and the figures are clean and easy to understand.
* As an application paper, the work demonstrates a nice implementation of combining a differentiable simulator (as the decoder) with a neural network (as the encoder). The authors also build a physical prototype (Figure 6) based on the output of the approach. It is highly unusual to take this extra step to build a physical prototype and should therefore be commended.
* The results successfully highlight how using the simulator ensures that the final design results in a pin-jointed structure where the resolved forces at the internal pins are at equilibrium.

**Weaknesses:**

* The motivation for this approach is not entirely clear. The design pipeline appears to start with a $\tilde{\mathbf{X}}$ and then produces a set of stiffness values. These stiffness values, along with the boundary conditions are then fed to the mechanical simulator to result in the final design (presumably, with the neural network’s stiffness values):
    * In practice, how does one come up with a design $\tilde{\mathbf{X}}$?
    * How does the simulator only require $\mathbf{p}$ and $\mathbf{b}$ for shape prediction? It seems like the simulator is missing information to build a shell with $N$ pins.
* The cost of amortization (i.e. the number of direct optimization runs to build the data set for training) does not seem to be mentioned as a limitation of the approach. Are there instances in the real world where amortization of the optimization routine would be needed? For example, with a training set, it seems like 100 x direct optimization is required to train the model. This overhead would need at least 100 runs of the proposed approach to make it well-motivated. In a real design pipeline, how often would it be expected to call such a function? It already runs in less than 2 seconds for cable-net towers and around 6 seconds for the masonry shell. Is this this wall-clock time a problem in practice? If so, the paper would benefit from this written explicitly.

Minor
* $p$ for norm and $\mathbf{p}$ for loads are a bit confusing in terms of notation.
* Suggestion: use test-time inference rather than inference for predictions, since inference can also refer to inference over parameters.

**Questions:**

* Are the authors going to be releasing the code? As an application paper, it would be beneficial to the community if they were.
* Does the chosen bar stiffness affect the loading on the design since a stiffer bar might lead to a heavier design?
* Could you provide further clarification on how the stress direction is determined a-priori?
* When training the neural network component, how is this loss accounting for permutation invariance with locations, or is there some implicit consistency in the ordering of the pin coordinates.
* What mechanism is in place to prevent the model providing excessively large stiffness values for all the bars?
* Could you explain why the masonry shell is modeled as a pin-jointed structure but is then physically built as a shell (and how the two structure types are related)?

---

> ### Author Response · Authors · 2024-11-22
>
> 1. **Are the authors going to be releasing the code?**
>
> Yes, we plan to release the code with the camera-ready version of the paper.
>
> 2. **Does the chosen bar stiffness affect the loading on the design since a stiffer bar might lead to a heavier design?**
>
> Not necessarily. While stiffness and mass are related concepts, the loading is independent of the stiffnesses in our experiments. The reason is that stiffness is a metric that quantifies the ratio between the force passing through a given bar ($f$) and the length of the bar ($l$). See Equation 8 in our updated submission. Therefore, a high stiffness means that either the force in a bar is high (potentially requiring a thicker bar if the material the bar is made of is weak), or simply that the bar is very short (small $l$).
>
> To avoid ambiguity, we define the magnitude of the loads upfront in our experiments. This is standard in engineering practice. In the shell task, for example, we set the magnitude to an area load of 0.5 force/area units (e.g., 0.5 kN / m2). This magnitude is commensurate with the loads in the materialized design generated by our model as a (tabletop) masonry vault of uniform thickness.
>
> 3. **Could you provide further clarification on how the stress direction is determined a-priori?**
>
> We set the stress direction by setting the signs of the stiffness values $\mathbf{q}$ that we input to the mechanical simulator. We define the signs of $\mathbf{q}$ in the last layer of our encoder with the direction vector $\mathbf{s}$ (see Equation 3): a negative entry $s$ prescribes compressive stress, and a positive factor, tensile stress.  We have added this clarification –-which was missing– to our updated submission, in L195-L196.
>
> It is worth emphasizing that we can set the signs in the stress response of a structure a priori because our choice of specialized mechanical simulator allows us to do so _for free_ (Appendix D). Defining the signs of the stress response a priori would be more significantly challenging with other traditional but still differentiable mechanical simulators, e.g. references [1] and [2], and would potentially involve equality constraints on the outputs of these other simulations. We mention this distinction in our description of the mechanical decoder in Section 2.3 in the updated submission.
>
> [1] G. Wu, “A framework for structural shape optimization based on automatic differentiation, the adjoint method and accelerated linear algebra,” Struct Multidisc Optim, vol. 66, no. 7, p. 151, Jun. 2023, doi: 10.1007/s00158-023-03601-0.
>
> [2] T. Xue et al., “JAX-FEM: A differentiable GPU-accelerated 3D finite element solver for automatic inverse design and mechanistic data science,” Computer Physics Communications, vol. 291, p. 108802, Oct. 2023, doi: 10.1016/j.cpc.2023.108802.
>
> 4. **When training the neural network component, how is this loss accounting for permutation invariance with locations, or is there some implicit consistency in the ordering of the pin coordinates.**
>
> Our shape loss does not encode any permutation invariance, it only focuses on pointwise distance minimization between nodes (L159 - L161). To maintain consistency in our predictions, we keep the same order of the vertices in the rows in $\mathbf{X}$ in each task, as the reviewer correctly pointed out.
>
> 6. **What mechanism is in place to prevent the model providing excessively large stiffness values for all the bars?**
>
> Our model has no explicit mechanism to limit the entries of $\mathbf{q}$. In the second task, we regularize training by adding a term to the loss function (Equation 6). Nevertheless, this regularization reduces the variance of $\mathbf{q}$ without imposing a specific upper bound. In direct optimization, we do enforce box constraints on stiffness values, capping $\mathbf{q}$ at 20. This information has been included in Appendix E of the revised manuscript.

---

> ### Author Response · Authors · 2024-11-22
>
> 6. **Could you explain why the masonry shell is modeled as a pin-jointed structure but is then physically built as a shell (and how the two structure types are related)?**
>
> Excellent question! That might not be intuitive at first, but there is a substantial body of research in mechanics behind this modeling approach, e.g. [1, 2, 3]. This body of research revolves around two main concepts: limit state analysis and the Heyman safe theorem.
>
> The behavior of masonry structures is governed by force equilibrium rather than material strength [1, 2]. This makes reasoning about the masonry behavior compatible with that of a pin-jointed bar system. The bar system hence represents a virtual network of compression forces flowing in the shell. This force network corresponds to one force state (among potentially many) in the masonry structure. If the force state satisfies the Heyman assumptions [2]; then the masonry structure is safe under the applied loads, and the solution corresponds to a lower bound of the collapse state (limit analysis).
>
> The Heyman assumptions are [2, 3]:
> - (i) Masonry’s compressive strength is inﬁnite,
> - (ii) masonry’s tensile strength is considered null, and
> - (iii) no sliding occurs between the masonry blocks of the structure.
>
> We have added a sentence to the paper at the start of Section 4.1 indicating these assumptions to the reader.
>
> [1] S. Huerta, “The Analysis of Masonry Architecture: A Historical Approach: To the memory of Professor Henry J. Cowan,” Architectural Science Review, vol. 51, no. 4, pp. 297–328, Dec. 2008, doi: 10.3763/asre.2008.5136.
> [2] J. Heyman, “The stone skeleton,” International Journal of Solids and Structures, vol. 2, no. 2, pp. 249–279, Apr. 1966, doi: 10.1016/0020-7683(66)90018-7.
> [3] R. Maia Avelino, A. Iannuzzo, T. Van Mele, and P. Block, “Assessing the safety of vaulted masonry structures using thrust network analysis,” Computers & Structures, vol. 257, p. 106647, Dec. 2021, doi: 10.1016/j.compstruc.2021.106647.
>
> 7. **In practice, how does one come up with a design $\hat{\mathbf{X}}$?**
>
> In practice, a design $\hat{\mathbf{X}}$ comes from an architect who proposes the global visual intent of a structure. Then, a structural engineer must counterpropose the architect’s proposal with $\mathbf{X}$, a mechanically efficient version of $\hat{\mathbf{X}}$. Broadly speaking, the level of happiness of the architect and the engineer is directly proportional to the distance between $\hat{\mathbf{X}}$ and $\mathbf{X}$. The closer the shapes are, the merrier. We envision the end users of our model to be both architects and engineers, facilitating the dialogue between both parties in a CAD environment at an accelerated pace (Appendix A).
>
> 8. **How does the simulator only require p and b for shape prediction? It seems like the simulator is missing information to build a shell with N pins.**
>
> The information to calculate geometry for a system with $N$ pins is embedded in the structure of the stiffness matrix $\mathbf{K} \in \mathbb{R}^{N \times N} $, which is created by our mechanical simulator. The stiffness matrix is assembled by the simulator based on the connectivity between the $N$ pins and the $M$ bars. Since we keep this discretization and the connectivity constant in each task, the assembly information remains fixed across our experiments ($\mathbf{q}$, $\mathbf{p}$, and $\mathbf{b}$, in contrast, change values per every shape input to our model).
>
> The initial submission included the equations to create the stiffness matrix, and we have now incorporated a more explicit description of the stiffness matrix assembly in Appendix D of our revised submission.

---

> > ### Author Response · Authors · 2024-11-22
> >
> > 9. **The cost of amortization (i.e. the number of direct optimization runs to build the data set for training) does not seem to be mentioned as a limitation of the approach. Are there instances in the real world where amortization of the optimization routine would be needed? For example, with a training set, it seems like 100 x direct optimization is required to train the model. This overhead would need at least 100 runs of the proposed approach to make it well-motivated. In a real design pipeline, how often would it be expected to call such a function? It already runs in less than 2 seconds for cable-net towers and around 6 seconds for the masonry shell. Is this this wall-clock time a problem in practice? If so, the paper would benefit from this written explicitly.**
> >
> > Thank you for your questions. In this paper, we do not build a labeled dataset of pairs $(\hat{\mathbf{X}}, \hat{\mathbf{q}})$ for training, which would require solving expensive optimization problems a priori, as the reviewer indicates. We only feed target shapes $\hat{\mathbf{X}}$ to our model which are computed on the fly at train time (e.g., Appendix F).
> >
> > Training our model is thus quick on a CPU: it only takes 140 seconds for the shells task and 810 seconds for the towers task Table 3). In relative terms, training the shell model equates to solving only 25 direct optimization problems in the same hardware. In Appendix B.3 of the updated submission, we perform new experiments where we observe that the number of direct optimization problems required to match the training time of our model decreases as the discretization of the structure increases (Table 3, Figure 14). The training time could be shortened in the future if we trained on GPUs.
> >
> > We anticipate that the training costs will be amortized during inference since we only need to train once per task to match potentially thousands of geometries within the same family of target shapes. We expect our model to be deployed as a plugin in commercial 3D modeling software, as we demonstrate in Appendix A, enabling real-time design exploration by multiple users working on the same task and parametrization. This would amount to multiple calls to our trained model, further justifying the low training effort.
> >
> > We motivate our work in Section 1 by identifying that direct optimization runtime is a bottleneck for design exploration (e.g., in L70-75 in the updated manuscript), which necessitates real-time solutions (in milliseconds) to optimization problems due to high-speed design cycles in practice. Our model provides a viable alternative to architects and engineers by generating such solutions three orders of magnitude faster than costly optimization, the current state-of-the-art.
> >
> > While we highlight amortization as one of the main benefits of our model, it is not the only one. As we demonstrate in the tower task, our model also provides superior initialization for direct optimization to arrive at better-performing designs with a faster convergence rate and without expensive human input (Section 4.2). That is another promising property that aims at further integrating neural networks into physical design.

---

> > > ### Comment · Reviewer_jNkr · 2024-11-26
> > > **Response**
> > >
> > > I thank the authors for their responses above. In particular, the additional table in the appendix (B.3) on training cost amortization helps motivate the approach. Just to check I understand it correctly, # Opt is the number of times you would need to use your surrogate approach to make the amortization cost worth it? If this is the case then the proposed approach would be potentially useful in practice. I will raise my score accordingly.

---

> > > > ### Author Response · Authors · 2024-11-26
> > > >
> > > > Yes, # Opts is the number of direct optimization problems to solve to justify training our model. Thank you for your feedback!

---

### Official Review · Reviewer_VHnA · 2024-11-03

**Soundness:** 3
**Presentation:** 3
**Contribution:** 2
**Rating:** 6
**Confidence:** 5

**Summary:**

This manuscript presents an interesting application of AE-type of network architecture to facilitate design and optimization of mechanical structures, mostly in (building) architectural structures. Although the method presented is mostly ad-hoc, the manuscript is well written, the proposed method appears sound, and the potential application is clearly indicated.

**Strengths:**

The paper is nicely written, although there are a few small details which could make the paper more understandable for neurips attendees, who may not have prior knowledge on the mechanical structure presented.

The method presented appears sound. The architecture as shown in Fig 1, leveraging JAX, makes intuitive sense.

**Weaknesses:**

Concerns to the paper:

1. Although it is not clearly explained, it appears that each trained model is closely related to the underlying architectural structure (masonry shells or cable-net tower). The question is whether the trained model is generalizable. Even for the shells, if $M$ (number of bars) and $N$ (the number of nodes) change, the model may have to be retrained. It's understandable that this does not reduce the practical value of the proposed method.

2. The runtime measurements have such large variances that they are not meaningful. The author(s) should consider changing the evaluation platform (macOS has many background processes and is not a good platform for benchmarking), and deploy better runtime measurement libraries.

3. In line 200 in Eq. (3), the fact that $\tau$ can be used to specify minimal stiffness is very interesting. But there are no more follow-up discussion on this topic.

4. Structure of the paper: the decoder is not discussed in the main body of the paper, mostly in the appendices. This need to be fixed.

Minor issues:
The terms *mechanical integrity* and *mechanical efficiency*  should be explained early in the paper to make it easier for readers to appreciate the paper.

**Questions:**

1. Does the term *fabrication requirements* imply that each bar in the shell examples has to follow additional constraints?
2. Provide more discussion on the usage of $\tau$ as pointed out in the previous section
3. Revise the manuscript to discuss the decoder in the main body of the manuscript.

Minor issues:
1. Define (or discuss) *mechanical integrity* and *mechanical efficiency* earlier in the manuscript. What do they exactly mean?

---

> ### Author Response · Authors · 2024-11-22
>
> 1. **Even for the shells, if  (number of bars) and  (the number of nodes) change, the model may have to be retrained. It's understandable that this does not reduce the practical value of the proposed method.**
>
> Our model would have to be retrained if the number of bars or nodes changes. This is because we currently use an MLP as the encoder, which fixes the dimensions of the input shape matrix X. However, we are not restricted to that encoder architecture since our model construction is general (L187-188).
>
> We state this limitation in Section 6.1 (L533-534) and point to looking at other encoder architectures in future works to generalize across multiple bar connectivities without retraining (L534-535). However, as the reviewer suggests, the limitation does not reduce the value of our work.
>
> To be more forward with the reader about our model’s limitations, we have reworded a pair of sentences in Section 6.1 in the updated version of our manuscript. We would also like to point out that training our model is reasonably fast (for example, 1.20 minutes in the shell task in Section 4.1), and that retraining would not be unreasonable if the connectivity changes to a moderate extent.
>
> 2. **The runtime measurements have such large variances that they are not meaningful. The author(s) should consider changing the evaluation platform (macOS has many background processes and is not a good platform for benchmarking), and deploy better runtime measurement libraries.**
>
> Thank you for catching this problem. We build on our response to Reviewer NVMt to a similar question. The large variances in both tasks arose from a bug in our timing protocol. This bug was independent of the operating system.
>
> Despite having pre-compiled the models, the just-in-time (JIT) compilation of JAX was re-triggered when we ran inference on the first shape in the test batch due to an inadvertent change in the batch dimensions. We were thus including the re-compilation time (in the 300-400 ms range) in our statistics, which drove the variances up.
>
> We have fixed this bug and re-timed test inference for our model and the neural baselines. We have updated tables 1 and 2 with the new statistics in both tasks. Regarding tooling, now we have put checks in our code to ensure that JIT re-compilation time does not leak into our statistics again (e.g., by using tools like `equinox.debug.assert_max_traces` and setting the max number of traces to 1).
>
> The variance is now in the submillisecond levels. After discarding the JIT compilation time, the NN and PINN inference times decreased by one order of magnitude, making for stronger baselines.
>
> 3. **In line 200 in Eq. (3), the fact that $\tau$ can be used to specify minimal stiffness is very interesting. But there are no more follow-up discussion on this topic. Provide more discussion on the usage of $\tau$.**
>
> Thank you for the suggestion. While we briefly mentioned in our original submission (L203-202) that setting $\tau>0$ is equivalent to setting a lower box constraint on the stiffness values like in traditional optimization, we agree that the paper benefits from more discussion. To this end, we have extended Section 2.3 in our updated submission and mention that lower bounding the stiffness space with $\tau$ is beneficial for two reasons. First, it is beneficial to guide learning towards particular solutions since the map from $\mathbf{q}$ to $\mathbf{X}$ is non-unique due to mechanical indeterminacy [1, 2]. That is, multiple distributions of $\mathbf{q}$ can map to the same $\mathbf{X}$. Second, it is beneficial to provide numerical stability when learning to amortize the design of more complex systems that mix tensile and compressive bar forces, as we do in Task 2 in the paper. We also now state the value of tau we use in the shells task.
>
> [1] T. Van Mele, L. Lachauer, M. Rippmann, and P. Block, “Geometry-Based Understanding of Structures,” Journal of the International Association for Shell and Spatial Structures, vol. 53, no. 4, pp. 285–295, Dec. 2012.
>
> [2] S. Pellegrino and C. R. Calladine, “Matrix analysis of statically and kinematically indeterminate frameworks,” International Journal of Solids and Structures, vol. 22, no. 4, pp. 409–428, Jan. 1986, doi: 10.1016/0020-7683(86)90014-4.

---

> > ### Author Response · Authors · 2024-11-22
> >
> > 4. **Does the term fabrication requirements imply that each bar in the shell examples has to follow additional constraints?**
> >
> > Yes, they could. Although we mention fabrication and geometric constraints in the introduction of our submission as motivators (L35-37), in this paper, we only focus on the latter by matching target geometries.
> >
> > 5. **Revise the manuscript to discuss the decoder in the main body of the manuscript.**
> >
> > We have created a dedicated paragraph in Section 2.3 that discusses the decoder at a high level. Some of the information previously in Section 2.1 is now found in this new paragraph. The details of the decoder have been extended and remain in Appendix D.  We believe the explicit mention of the decoder in the main body of the manuscript enhances readability, as the reviewer pointed out. Thanks for the suggestion.
> >
> > 6. **Define (or discuss) mechanical integrity and mechanical efficiency earlier in the manuscript.**
> >
> > To clarify these terms, we have delineated more clearly what we mean by mechanical efficiency (L34-L35) and mechanical integrity (L84-L85) in Section 1 of our updated submission. In sum, by mechanical efficiency, we mean sustaining loads with low strain energy and thus with low material volume (i.e. material minimization); by mechanical integrity, we mean the accuracy in predicting the mechanical response of a structure by respecting physics.

---

> > > ### Comment · Reviewer_VHnA · 2024-11-25
> > > **Path to a practical surrogate model**
> > >
> > > The revision addressed my concerns. The paper is now more understandable. The paper provides a path to a surrogate model with practical application.
> > >
> > > I have revised my rating accordingly.

---

> > > > ### Author Response · Authors · 2024-11-25
> > > >
> > > > Thank you for looking at our updated submission!

---

### Official Review · Reviewer_NVMt · 2024-11-05

**Soundness:** 3
**Presentation:** 2
**Contribution:** 2
**Rating:** 6
**Confidence:** 4

**Summary:**

The paper introduces a physics-in-the-loop scheme for the design of architectural structures. The authors here use a neural network(MLP) to learn the mapping from the desired structural shape to intermediate mechanical properties (bar stiffnesses) and then use a physics model to generate a physically feasible shape approximating the target.  The authors then apply this method to the design of masonry shells and cable towers,  comparing it with two neural network baselines: one trained to produce physically feasible shapes and the other trained to produce feasible shapes while also ensuring mechanical stability. Results on both these case studies show the proposed approach outperforming both these baselines and being on par with numerical optimization.

**Strengths:**

1. The idea of combining physics with ML for architectural design is promising, as it removes the need for generating costly training labels. Instead, the model can learn by integrating the physics model with inexpensive loss functions.
2. The out-of-distribution performance is also interesting, as it potentially reduces the need for extensive variability in the training data.

**Weaknesses:**

1. The case studies provided are relatively simplistic and do not reflect real-world applications. The physics model used is also quite basic, limiting the method's applicability in practical scenarios.
2. Additionally, as the authors acknowledge, even when trained, the proposed parameterization lacks flexibility and requires retraining whenever the design representation changes.

**Questions:**

Overall, the paper is technically sound and proposes an interesting integration of ML and physics. However, there are some concerns as follows:

1. The proposed hybrid approach requires running the physics model during inference, likely increasing training costs. While the authors report inference time, the training time is not provided. Given the limited representational capacity of the network, it must be retrained whenever the design is re-parameterized (a likely scenario during conceptualization). Reporting training times would enable a better assessment of the method’s real-world viability; if training significantly exceeds the duration of several optimizations, direct optimization might be preferable.

2. Similarly, providing training time metrics for the other baselines would help clarify trade-offs and be useful in scenarios where slight accuracy losses are acceptable for performance gains.

3. From Fig.9 , the optimization initialized with the proposed method converges quickly as compared to the other initializations. This leads to an interesting question of how the NN initialized optimizations would perform. This approach would have the benefit of outputting a guaranteed local minima while avoiding the additional physics overhead during training. Including these results would enhance the paper's contribution to the community.

4. The case studies considered here have relatively low DOFs, and the physics relies on a linear FDM. The authors could discuss the viability of this approach for structures where linear FDM is inapplicable or for dynamic scenarios. In such cases—and even when FDM is applicable but the structure has much higher DOFs—would this approach remain feasible?

5. Furthermore, the variation in MLP and PINN inference times is puzzling. Since the input sizes remain constant, one would expect the inference times for the fully trained networks to be similar. Could the authors comment on this discrepancy?

---

> ### Author Response · Authors · 2024-11-22
>
> 1. **Reporting training times would enable a better assessment of the method’s real-world viability; if training significantly exceeds the duration of several optimizations, direct optimization might be preferable.**
>
> We reported the training times of our model in Table 3 in the Appendix in our original submission (now Table 4). The training time of our model is in the order of a few minutes per task on CPU (140 seconds for the shells task and 810 for the towers task). Regarding viability, for example, the training time of our model is equivalent to solving 25 direct optimization problems in sequence in the shells task (taking 5.81 seconds as the average for optimization, Table 1). Once trained, our model allows for many more shell design iterations than 25.
>
> 2. **Similarly, providing training time metrics for the other baselines would help clarify trade-offs and be useful in scenarios where slight accuracy losses are acceptable for performance gains.**
>
> We also provided training times for the neural baselines (NN and PINN) in Table 3 in the Appendix of our first submission (now Table 4). The training times of the baselines are in the order of minutes on a CPU but are shorter than our model because they do not run the mechanical simulation during training. We note that the difference in training times varies with the task complexity: in the shell task, the difference between the baselines and our model is only 1 minute, and in the cable net task (a more challenging task), that number rises to 11 minutes. While the NN and the PINN are faster to train, neither of them drives the physics loss to zero, which is one of the key drivers in our paper. Although our model is more expensive to train than the purely neural baselines, it satisfies the physics objective by construction and compensates for the training time by generating fast physics-constrained design iterations in inference in real time.
>
> 3. **From Fig.9 , the optimization initialized with the proposed method converges quickly as compared to the other initializations. This leads to an interesting question of how the NN initialized optimizations would perform. This approach would have the benefit of outputting a guaranteed local minima while avoiding the additional physics overhead during training. Including these results would enhance the paper's contribution to the community.**
>
> Great idea! We performed an additional optimization experiment in the towers task, utilizing the PINN predictions as the initialization. We show the results in the main text and the updated version of Figure 9. Just like our model, the PINN initializations converge faster than expert initialization and arrive at better-performing designs. Nevertheless, the PINN initialization is slower to converge than our model and only reaches our model’s shape loss at the end of the convergence curve.

---

> > ### Author Response · Authors · 2024-11-22
> >
> > 4. **The case studies considered here have relatively low DOFs, and the physics relies on a linear FDM. The authors could discuss the viability of this approach for structures where linear FDM is inapplicable or for dynamic scenarios. In such cases—and even when FDM is applicable but the structure has much higher DOFs—would this approach remain feasible?**
> >
> > Our method remains feasible for structures that can be modeled as pin-jointed bar systems and that have a higher DOF count.
> >
> > We performed an additional set of experiments in the shell task to this end, where we increased the number of bars in the problem from $M=180$ to $M=1012$. The results are in Appendix B.3 of the updated manuscript. The training cost and the test inference time increase with the problem size, but the predictions remain adequate fits to the target shapes (see Figure 14). This demonstrates our method works in tasks with much higher DOFs than what we presented in Section 4.1 of our original submission.
> >
> > Comparing our model with direct optimization on problems with more DOFs amplifies its value. While in the problems we present in Section 4.1 the expected optimization runtime to match one shape is 5.8 seconds ($M=180$), the optimization runtime increases by two orders of magnitude when $M=480$. If we measure the training cost of our model by the number of direct optimization problems a user would solve in parallel, it would only take 7 direct optimization runs to justify training when $M=180$, instead of 25. This training-to-optimization ratio can further decrease from 7 to 2 when $M=1012$.
> >
> > 5. **Furthermore, the variation in MLP and PINN inference times is puzzling. Since the input sizes remain constant, one would expect the inference times for the fully trained networks to be similar. Could the authors comment on this discrepancy?**
> >
> > We understand the reviewer refers to the variance in the inference times. The large variances in both tasks arose from a bug in our timing protocol. Despite having pre-compiled the models, the just-in-time (JIT) compilation of JAX was re-triggered when we ran inference on the first shape in the test batch due to an inadvertent change in the batch dimensions. We were thus including the re-compilation time (in the 300-400 ms range) in our statistics, which drove the variances up.
> >
> > We fixed this bug and re-timed test inference for our model and the neural baselines. We have updated tables 1 and 2 with the new statistics in both tasks. The variance is now in the submillisecond levels, and the NN and PINN times are equal. After discarding the JIT compilation time, the NN and PINN inference times decreased by one order of magnitude, making for stronger baselines. In the shell task, our model performance improved too, further increasing the gap with direct optimization.

---

> > > ### Comment · Reviewer_NVMt · 2024-11-27
> > > **High-DOF Results Strengthen Motivation for Practical Applications**
> > >
> > > The authors have thoroughly addressed my concerns. The updated computation times and optimization results effectively demonstrate the overall performance of the proposed approach. As noted by other reviewers, this provides a clear path toward developing a practical surrogate model for pin-jointed bar structures.

---

> > > > ### Author Response · Authors · 2024-11-29
> > > >
> > > > Thank you! We are glad to hear that our responses have adequately addressed the reviewer's questions and that our new experiments demonstrate the feasibility of our model. In light of our revised work, would the reviewer consider updating their score?

---

### Author Response · Authors · 2024-11-22
**General response**

We thank all reviewers for their time and feedback on our submission.

We are encouraged by the reviewers’ commendation for building a physical prototype based on our model predictions (jNKr, H1FF). We are also motivated because they found our work valuable as it removes the need for costly training labels and has good out-of-distribution performance (NVMt); and because it allows setting a lower bound on the inputs to the decoder (VHnA). The reviewers found the manuscript was well-written (VHnA), well-presented (H1FF, jNKr), and technically sound (VHnA, NVMt).

To address the reviewer’s comments, we have uploaded a revised version of our manuscript. For convenience, we explicitly track modifications in our resubmission by coloring text changes in blue. We list the main changes below and respond to each of the reviewer’s comments individually.

- Added a new experiment in the towers task, where we use the PINN predictions as initialization for direct optimization. We updated Figure 9 to include the convergence curve of this initialization (NVMt).
- Added a new set of experiments in Appendix B.3 where we quantify trade-offs between our model and direct optimization in terms of training vs inference time in problems with up to 1000 DOFs (NVMt).
- Included a subsection in Section 2.3 of the main text that describes the mechanical decoder at a high level. We also extended Appendix D to better explain the minutiae of how our mechanical decoder works (VHnA).
- Fixed a bug in our timing protocol that increased the variance of the test inference times. The large variances stemmed from mistakenly including the JIT compilation time in our statistics. We have updated Tables 1 and 2 accordingly. The updated inference times make for stronger neural baselines, but the benefits of our model w.r.t. optimization in terms of inference time are preserved (NVMt, VHnA).
- Defined and distinguished between mechanical efficiency and integrity (VHnA).
- Described what mechanical theory allows us to model masonry shells as pin-jointed bar systems (jNKr).
- Updated Figures 1 and 3 to enhance legibility (H1FF).
- Changed the term “bar stress” to “bar force” to be consistent with the information our mechanical simulator outputs.
- Shortened the abstract to meet page limits.
- Fixed typos.

---

### Meta-Review · Area_Chair_sUsz · 2024-12-24

**Metareview:**

The paper introduces a surrogate model for designing architectural shape structures modeled as bar systems, a simplified structural representation consisting of nodes connected by bars. This model combines a neural network (NN) with a differentiable mechanical simulator. Given an input target shape, provided, for example, by a designer, the NN computes stiffness parameters that encode the mechanical behavior of the structure, while the simulator outputs the corresponding structure. The model is trained end-to-end. Validation is performed on two problems: masonry shells and cable-net towers. The baselines consist of an encoder-decoder NN, the same architecture augmented with an additional mechanical loss (similar to a physics-informed neural networks (PINNs) approach), and a traditional direct optimization. The proposed hybrid model approaches the performance of direct simulation methods at a much lower computational cost.

This is an application paper, and the idea of combining differentiable physical models with NNs has been explored in various fields. The originality of this work lies in its application domain. The reviewers consider this work, even though it is evaluated on relatively simple mechanical design problems, to be a significant first step toward developing practical applications. The authors’ responses and the additional experiments provided during the rebuttal adequately address the reviewers' concerns. I recommend accepting the paper.

**Additional Comments On Reviewer Discussion:**

All the reviewers were satisfied with the clarifications provided by the authors during the rebuttal and felt that their main concerns had been addressed.

---

### Decision · Program_Chairs · 2025-01-22

Accept (Poster)